# The inner junction complex of the cilia is an interaction hub that involves tubulin post-translational modifications

Ahmad Abdelzaher Zaki Khalifa[1,2†], Muneyoshi Ichikawa[1†‡], Daniel Dai[1,2], Shintaroh Kubo[2,3], Corbin Steven Black[1,2], Katya Peri[1], Thomas S McAlear[1,2], Simon Veyron[2,4], Shun Kai Yang[1,2], Javier Vargas[1,2], Susanne Bechstedt[1,2], Jean-François Trempe[2,4], Khanh Huy Bui[1,2]*

[1]Department of Anatomy and Cell Biology, McGill University, Québec, Canada; [2]Centre de Recherche en Biologie Structurale - FRQS, McGill University, Québec, Canada; [3]Department of Biophysics, Graduate School of Science, Kyoto University, Kyoto, Japan; [4]Department of Pharmacology & Therapeutics, McGill University, Montréal, Canada

*For correspondence:
huy.bui@mcgill.ca

†These authors contributed equally to this work

Present address: ‡Department of Systems Biology, Graduate School of Biological Sciences, Nara Institute of Science and Technology, Nara, Japan

Competing interests: The authors declare that no competing interests exist.

**Abstract** Microtubules are cytoskeletal structures involved in stability, transport and organization in the cell. The building blocks, the α- and β-tubulin heterodimers, form protofilaments that associate laterally into the hollow microtubule. Microtubule also exists as highly stable doublet microtubules in the cilia where stability is needed for ciliary beating and function. The doublet microtubule maintains its stability through interactions at its inner and outer junctions where its A- and B-tubules meet. Here, using cryo-electron microscopy, bioinformatics and mass spectrometry of the doublets of *Chlamydomonas reinhardtii* and *Tetrahymena thermophila*, we identified two new inner junction proteins, FAP276 and FAP106, and an inner junction-associated protein, FAP126, thus presenting the complete answer to the inner junction identity and localization. Our structural study of the doublets shows that the inner junction serves as an interaction hub that involves tubulin post-translational modifications. These interactions contribute to the stability of the doublet and hence, normal ciliary motility.

## Introduction

Cilia and flagella are highly conserved organelles present in protists all the way to humans. They are commonly classified into two forms: motile and non-motile cilia. Motile cilia are responsible for mucus clearance in the airway, cerebrospinal fluid circulation and sperm motility (*Satir and Christensen, 2007*). The non-motile cilia, namely primary cilia, function as the cellular antennas that sense chemical and mechanical changes. Cilia are essential for growth and development and therefore human health. Defects in cilia often result in abnormal motility or stability, which lead to cilia-related diseases such as primary ciliary dyskinesia, retinal degeneration, hydrocephalus and polydactyly (*Hurd and Hildebrandt, 2011*).

Both cilia types are comprised of a bundle of nine specialized microtubule structures termed doublet microtubules (doublets). Ciliary components, important for motility such as the outer and inner dynein arms, radial spokes and the dynein regulatory complex (DRC) are assembled onto the surface of the doublet (*Bui et al., 2008*; *Bui et al., 2009*; *Bui et al., 2012*; *Heuser et al., 2009*). Inside the doublets, is a weaving network of proteins, termed microtubule-inner-proteins (MIPs), that bind to the inner lumen surface of the doublet (*Ichikawa et al., 2017*; *Ichikawa et al., 2019*). These MIPs act to stabilize the doublet and possibly regulate the ciliary waveform through interactions with the tubulin lattice (*Ichikawa et al., 2019*).

Doublets consist of a complete 13-protofilament (PF) A-tubule, similar to a 13-PF cytoplasmic microtubule and a partial 10-PF B-tubule forming on top of the A-tubule (naming of the PF number is shown in *Figure 1B*). To this day there still exists a long-standing question of how the junctions between the two tubules are formed (*Tilney et al., 1973*; *Linck and Stephens, 2007*; *Nicastro et al., 2011*). The recent high-resolution cryo-EM structure of the doublet shows that the outer junction is formed by a non-canonical tubulin interaction among PF B1, A10 and A11 (*Ichikawa et al., 2017*). The inner junction (IJ), which bridges the inner gap between the B-tubule

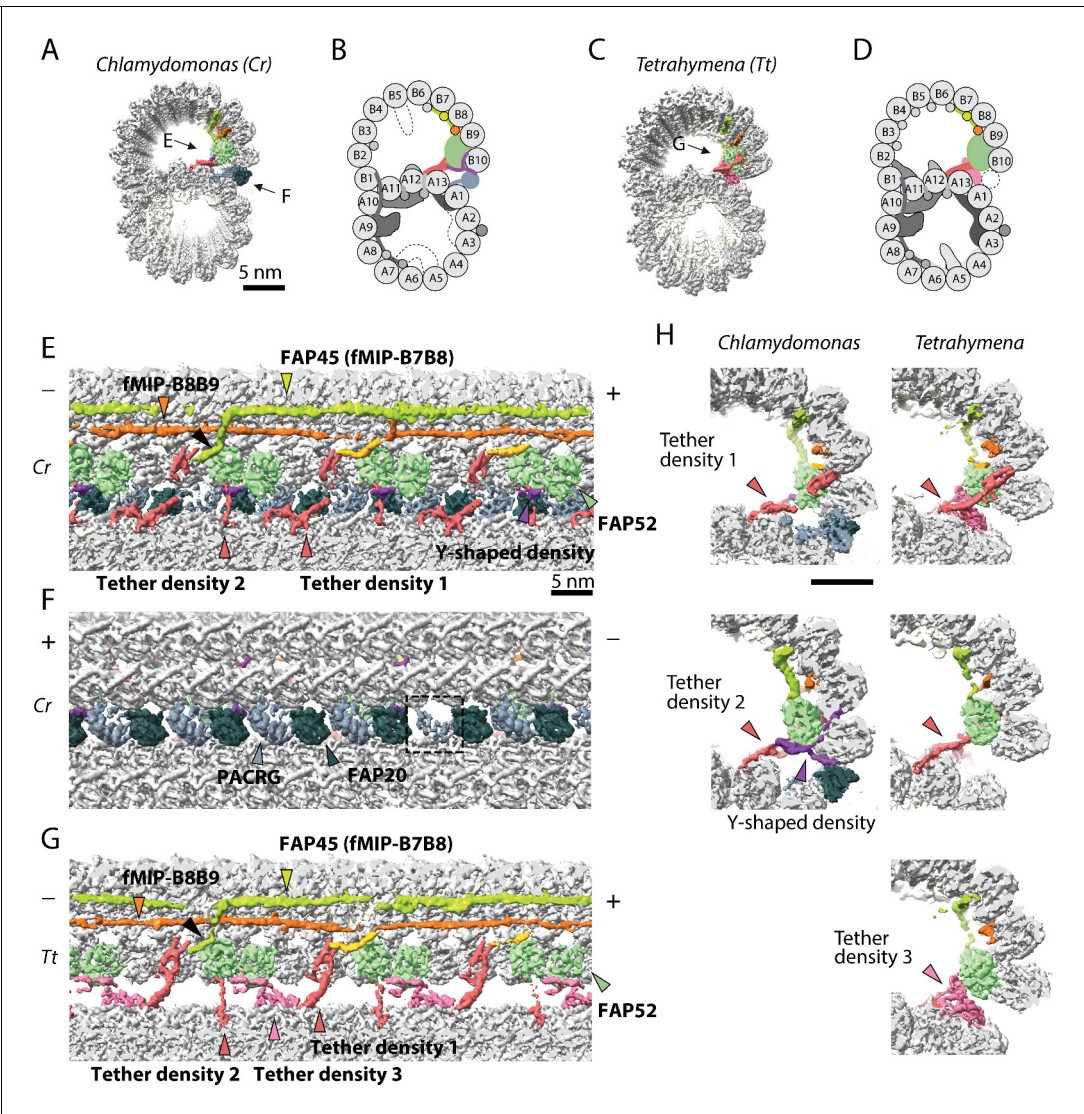

**Figure 1.** The IJ structures of *Chlamydomonas* and *Tetrahymena* doublet. (**A–D**) Surface renderings and schematics of the 48 nm repeat cryo-EM maps of *Chlamydomonas* (**A, B**) and *Tetrahymena* (**C, D**) doublets viewed from the tip of the cilia. Black arrow indicates longitudinal view in (**E**), (**F**) and (**G**). (**E–F**) The longitudinal section of the *Chlamydomonas* doublet at the IJ complex from the inside (**E**) and outside (**F**). (**G**) The longitudinal section of *Tetrahymena* doublet viewed from the inside. Color scheme: FAP20: dark green; PACRG: gray; FAP52: light green, Y-shaped density: purple; FAP45: yellow green; fMIP-B8B9: orange; Tubulin: light gray; Unknown density: yellow; Rest of MIPs: white; Tether density 1 and 2: red; Tether density 3, pink. Plus and minus ends are indicated by + and - signs. (**H**) Cross sectional views of the different Tether densities from *Chlamydomonas* (left) and *Tetrahymena* (right). In *Chlamydomonas*, there is a Y-shaped density (purple) that cradles the FAP52 density. The Y-shaped density is absent in *Tetrahymena*. In *Tetrahymena*, we observed Tether density 3, which is absent in *Chlamydomonas*.

The online version of this article includes the following figure supplement(s) for figure 1:

**Figure supplement 1.** Data related to the doublet structures.

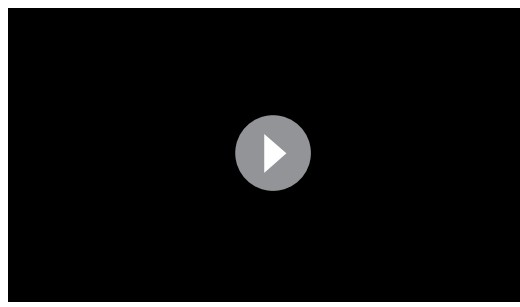

**Video 1.** The IJ complex of the *Chlamydomonas reinhardtii* flagella.
https://elifesciences.org/articles/52760#video1

and A-tubule is formed by non-tubulin proteins. Both primary and motile cilia have been observed to contain the IJ (*Nicastro et al., 2011*; *Sun et al., 2019*).

In vitro formation of a B-tubule-like hook (i.e. the outer junction like interaction) was assembled onto pre-existing axonemal and mitotic spindle microtubule with the addition of purified brain tubulin (*Euteneuer and McIntosh, 1980*). More recently, the B-tubule-like hook can be achieved by adding purified tubulins onto existing subtilisin-treated microtubules (*Schmidt-Cernohorska et al., 2019*). However, these hooks are not closed and appear to be flexible (*Schmidt-Cernohorska et al., 2019*). This supports the notion that the IJ is composed of non-tubulin proteins that are indispensable to the stability of the IJ.

The IJ is composed of FAP20 as shown through cryo-electron tomography (*Yanagisawa et al., 2014*). *Dymek et al. (2019)* reported that PArkin Co-Regulated Gene (PACRG) and FAP20 proteins form the IJ. PACRG and FAP20 are arranged in an alternating pattern to form the IJ linking the PF A1 of A-tubule and PF B10 of the B-tubule. In addition, both FAP20 and PACRG are important components for motility (*Yanagisawa et al., 2014*; *Dymek et al., 2019*). Both PACRG and FAP20 are conserved among organisms with cilia, suggesting a common IJ between species.

PACRG shares a bi-directional promoter with the Parkinson's disease-related gene parkin (*Kitada et al., 1998*; *West et al., 2003*). Knockdowns of PACRG genes in *Trypanosoma brucei* and *Xenopus*, lead to defects in the doublet structure and, therefore, impaired flagellar motility. In vertebrates, defects in left-right body symmetry and neural tube closure were observed from knockdowns of PACRG (*Thumberger et al., 2012*). In mice, PACRG knockout results in male sterility (*Lorenzetti et al., 2004*) and hydrocephalus (*Wilson et al., 2010*). FAP20 knockout mutants in *Chlamydomonas* have motility defects and frequent splaying of the axoneme (*Yanagisawa et al., 2014*). Similarly, FAP20 knockdown in *Paramecium* has an altered waveform (*Laligné et al., 2010*). A recent report identified other MIPs near the IJ, namely FAP52 and FAP45 (*Owa et al., 2019*). Knockouts of FAP52 or FAP45 lead to an unstable B-tubule in *Chlamydomonas*. Double knockouts of FAP52 or FAP45 together with FAP20 lead to severe damage of the B-tubule. The gene deletion of the human homolog of FAP52 has been shown to cause heterotaxy and *situs inversus totalis* in patients (*Ta-Shma et al., 2015*).

Cryo-EM structures of isolated doublets from *Tetrahymena* show that there are different tethering densities that connect the B-tubule to the A-tubule aside from the IJ (*Ichikawa et al., 2017*; *Ichikawa et al., 2019*). However, the identity of such protein remains unknown to date. Taken together, these data suggest that there is a complex interaction at the IJ region involving multiple proteins in addition to PACRG, FAP20, FAP45 and FAP52. These interactions may play a role in regulating ciliary motility via stability.

Despite all the phenotypes known about these IJ proteins, there are not yet any high-resolution structures to explain the molecular mechanism of the B-tubule closure and the IJ stability. In this study, we present the high-resolution cryo-EM structure of the IJ region from the *Chlamydomonas* doublet. Using a combination of bioinformatics and mass spectrometry, we were able to identify two new IJ proteins, FAP276 and FAP106, and a new IJ-associated MIP, FAP126. Our results suggest that the IJ is made up of a complex of proteins involving PACRG, FAP20, FAP52, FAP276, FAP106 and FAP126. We also compare the *Chlamydomonas* structure with the *Tetrahymena* structure to understand the common and species-specific features of the IJ.

## Results

### Multiple tether proteins exist at the IJ

We obtained the 48 nm repeating unit of taxol stabilized and salt-treated *Chlamydomonas* doublet at 4.5 Å resolution (*Figure 1A,B* and *Figure 1—figure supplement 1A–C*). Due to the salt wash, some MIPs were lost when compared to the intact tomographic doublet structure (dashed parts in *Figure 1B*) (*Bui et al., 2012*). On the other hand, the IJ region bridging PF B10 and A1 of the *Chlamydomonas* doublet remained intact (*Figure 1A,B*). In the corresponding salt-treated *Tetrahymena* doublet map, most of the IJ region bridging PF B10 and A1 is missing (*Ichikawa et al., 2017*; *Ichikawa et al., 2019*) (*Figure 1C,D*, more details later). Based on previous studies (*Yanagisawa et al., 2014*; *Dymek et al., 2019*; *Owa et al., 2019*), we were able to localize FAP52, FAP45 in both *Tetrahymena* and *Chlamydomonas* (FAP52, light green and FAP45, yellow-green in *Figure 1E,G*), and PACRG and FAP20 (PACRG, light gray and FAP20, dark gray in *Figure 1F*) in *Chlamydomonas*.

In this study, we termed the structure formed by the repeating units of PACRG and FAP20, the IJ protofilament (IJ PF), and refer to the IJ complex as all the proteins involved in the attachment of the B-tubule to the A-tubule. Most of the proteins in this IJ complex are attached to PFs B8 to B10 and the IJ PF.

The presence of the complete IJ PF stabilizes the B-tubule of the *Chlamydomonas* doublet compared to that of *Tetrahymena*, as evidenced by local resolution measurements (*Figure 1—figure supplement 1D*). Despite having a good resolution in the A-tubule, the *Tetrahymena* doublet has a significantly lower resolution in the IJ area of the B-tubule.

Inside the B-tubule of both species, it is clear that the IJ region is held up by many tether densities along the doublet connecting the B-tubule to PF A13 (*Figure 1E–G*). First, the B-tubule is held up by Tether density 1 (red, *Figure 1H*), referred to as MIP3b previously (*Ichikawa et al., 2017*; *Ichikawa et al., 2019*). Tether density 1 connects the PF B9/B10 and A13. The second connection is named Tether density 2 (red, *Figure 1E–G*), projecting from the proximal lobe of the FAP52 density (referred to as MIP3a previously *Ichikawa et al., 2017*) and connecting to PF A13 (*Figure 1H*). In *Chlamydomonas*, there is another Y-shaped density (purple) cradling the FAP52 proximal lobe and projecting into the gap between the IJ PF and PF B10 (*Figure 1H*).

FAP45 is referred to as MIP3c previously (*Owa et al., 2019*). Cryo-electron tomography of *Chlamydomonas* FAP45 mutant indicate it is a L-shaped filamentous MIP binding at the inner ridge between PF B7 and B8 (*Figure 1E*) (*Owa et al., 2019*). In both species, the L-shaped FAP45 contacts FAP52 once every 48 nm. This explains the pull-down of FAP45 using FAP52 antibody and vice versa after chemically crosslinking the *Chlamydomonas* axonemes by a zero-length cross-linker (*Owa et al., 2019*). In *Tetrahymena*, there exists a Tether density 3 (pink, *Figure 1G,H*), projecting from the distal lobe of the FAP52 density and connecting to PF A13. This Tether density 3 is not present in the *Chlamydomonas* doublet, suggesting that this density is specific to *Tetrahymena*. All the tether densities described above repeat with 16 nm.

The IJ PF is formed by a pair of PACRG and FAP20 repeating every 8 nm with the same repeating unit as tubulin dimers (*Figure 1F*). This is to be expected as the purpose of the IJ PF is to bridge the tubulin dimers from PFs B10 and A1. In the 48 nm *Chlamydomonas* doublet map, there is one PACRG unit with a less defined density compared to the others (dashed box, *Figure 1F*). It has been shown that there is one PACRG density missing in every 96 nm repeat (*Heuser et al., 2009*; *Dymek et al., 2019*). Since our doublet map is a 48 nm repeat unit, the less defined density of PACRG corresponds to the average from one unit of PACRG and one missing unit, that is half the signal. This missing unit of PACRG in the 96 nm repeat allows the basal region of the DRC to anchor onto the doublet (*Heuser et al., 2009*) (*Figure 1—figure supplement 1F,G*).

The entire IJ filament of PACRG and FAP20 is previously reported missing in the *Tetrahymena* structure, probably due to salt wash and dialysis treatment (*Ichikawa et al., 2019*). However, upon adjusting the threshold value of the surface rendering, we observed one pair of PACRG and FAP20 remaining in the structure (*Figure 1—figure supplement 1E*) (*Ichikawa et al., 2017*). This can be a result of a specific region in the 96 nm repeat of the *Tetrahymena* doublet that has extra interactions to prevent this pair of PACRG and FAP20 to detach during sample preparation.

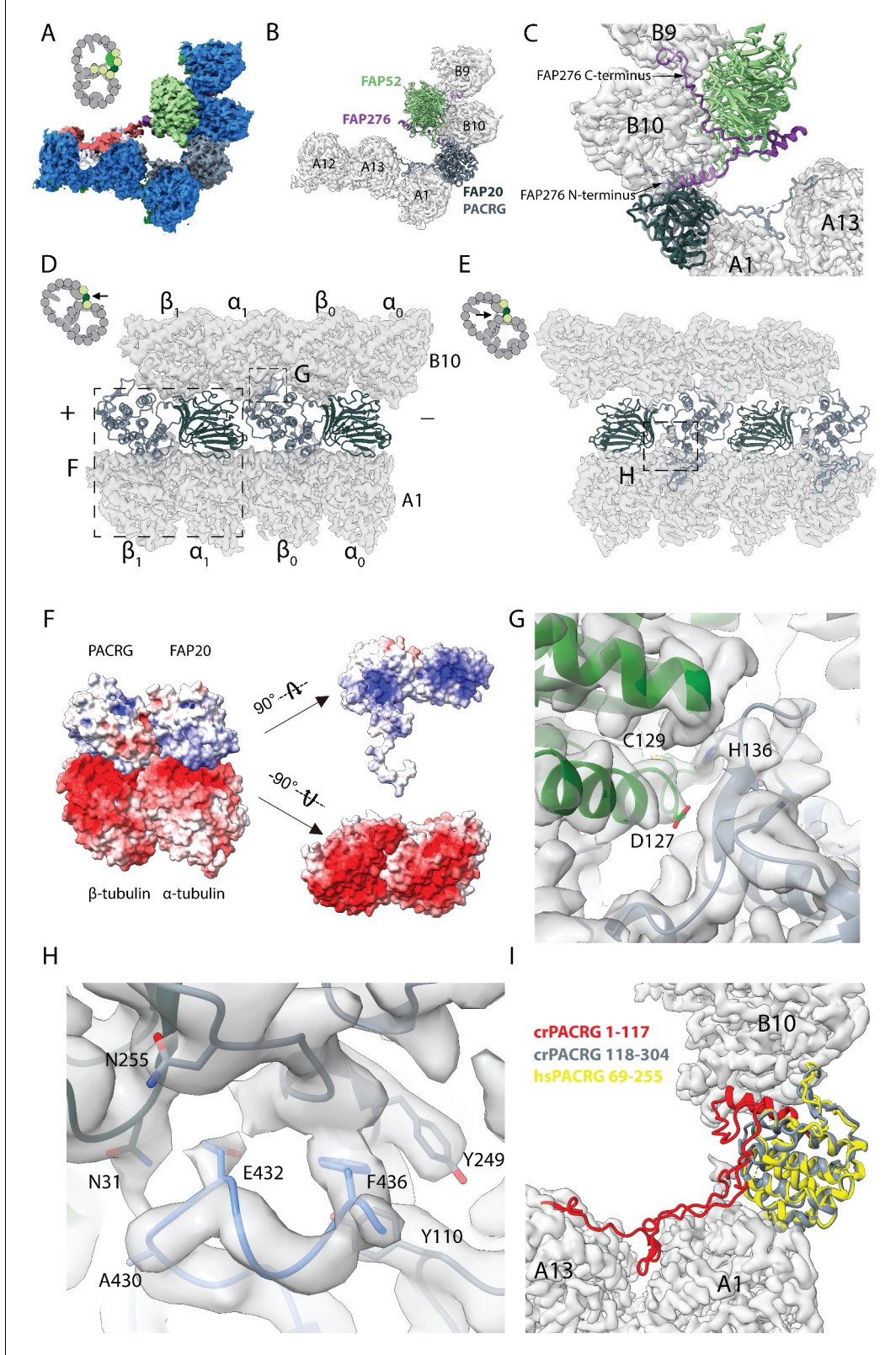

**Figure 2.** 16 nm structure of *Chlamydomonas* doublet. (**A, B**) 16 nm repeat structure of *Chlamydomonas* doublet and model at the IJ region. Tubulin densities are depicted as transparent gray, except in A in which tubulins are shown in dark blue. (**C**) Atomic model of the IJ complex, consisting of PACRG, FAP20, FAP52 and FAP276. Color scheme: FAP20: dark green; PACRG: gray; FAP52: light green, FAP276: purple; tubulin: transparent gray density. *Figure 2 continued on next page*

*Figure 2 continued*

(D, E) Maps and models of PF A1 and B10, and IJ PF illustrating how the IJ PF interacts with tubulins. The views are indicated in the schematics. Dashed boxes indicate the views in (F), (G) and (H). Color scheme: α-tubulin: green; β-tubulin: blue; PACRG: gray; FAP20: dark green; FAP276: purple. (F) Electrostatic surface charge of PACRG, FAP20 and α- and β-tubulins of PF A1. Tubulin surface is negatively charged while the interacting interface of PACRG and FAP20 are positively charged. (G) The interaction of the PACRG with the inter-dimer interface of tubulins from PF B10 is shown. H136 from PACRG is conserved and is likely to take part in the interaction with D127 and C129 from α-tubulin of PF B10. (H) The C-terminus of β-tubulin of PF A1 interacts with PACRG and FAP20. Potential residues involved in the interaction of C-tail of β-tubulin and PACRG and FAP20 are shown. (I) *Chlamydomonas* PACRG has a long N-terminus 1–117. The N-terminus of *Chlamydomonas* PACRG (red) forms a stable triple helix arrangement with the core of the protein. This is not observed in the human PACRG (PDB: 6NDU) shown in yellow. In addition, the N-terminus of PACRG going into the wedge between PF A13 and A1.

The online version of this article includes the following figure supplement(s) for figure 2:

**Figure supplement 1.** Atomic models of PACRG, FAP20, FAP52 and FAP276.

## PACRG, FAP20, FAP52 and FAP276 form an IJ complex

Since the majority of IJ proteins repeat with 8 nm and 16 nm, we first obtained the 16 nm repeating unit from *Chlamydomonas* and *Tetrahymena* at 3.9 Å resolution (*Figure 1—figure supplement 1C*). Using focused refinement, the IJ complex of *Chlamydomonas* was improved to 3.6 Å resolution (*Figure 2A,B*). Since the B-tubule is flexible in *Tetrahymena*, the resolution in the IJ area was significantly lower than that of *Chlamydomonas* (*Figure 1—figure supplement 1D*). In contrast, the IJ region of *Chlamydomonas* has good resolution with the intact IJ PF. It is worth mentioning that PFs A3-A6 in *Chlamydomonas* have lower resolution due to the loss of MIPs in this region. At 3.6 Å resolution, we were able to segment, trace and de novo model PACRG, FAP20 and FAP52 in *Chlamydomonas* (*Figure 2A,B* and *Figure 2—figure supplement 1A–F*). We could not model FAP45 since FAP45 repeats with 48 nm, and therefore was averaged out in the 16 nm averaged map.

The Y-shaped density is repeating with 16 nm and has a large binding interface with FAP52, we hypothesized that this protein would be missing in FAP52 knockout cells. Therefore, we did mass spectrometry of split doublets isolated from *Chlamydomonas* FAP52 knockout cells and performed relative quantification of axonemal proteins compared to the wild type (*Dai et al., 2019*). We observed 12 proteins completely missing (no peptide detected) and 26 proteins reduced by at least

**Table 1.** Proteins completely missing (no peptide detected) in FAP52 knockout mutant

| Names | Mol. weight (kDa) | p-values (WT vs FAP52) | Exclusive unique peptide counts in WT (quantitative values after normalization) |
|---|---|---|---|
| ARL3 | 20 | 0.0013 | 2, 2, 1 (2, 1, 2) |
| CHLREDRAFT_171815 | 57 | 0.035 | 2, 6, 1 (2, 5, 2) |
| CHLREDRAFT_156073 | 11 | 0.024 | 1, 1, 1 (2, 1, 2) |
| FAP276 | 10 | 0.015 | 3, 3, 2 (8, 7, 14) |
| FAP52 | 66 | 0.0046 | 27, 21, 15 (59, 73, 105) |
| FAP36 | 41 | 0.0023 | 3, 3, 1 (3, 3, 2) |
| CrCDPK1 | 54 | <0.0001 | 3, 5, 2 (3, 4, 4) |
| CHLREDRAFT_176830 | 110 | 0.024 | 2, 1, 2 (2, 1, 1) |
| FAP173 | 33 | 0.012 | 3, 3, 1 (5, 3, 2) |
| FAP29 | 112 | 0.00045 | 2, 3, 2 (3, 3, 4) |
| CHLREDRAFT_181390 | 41 | 0.028 | 1, 1, 1 (1, 1, 2) |
| ANK2 | 60 | 0,0041 | 1, 2, 1 (1, 1, 2) |

twofold (*Table 1* and Supplementary Table 1). PACRG, FAP20 and FAP45 levels are unchanged in FAP52 mutants since their binding interfaces with FAP52 are not as large as supported by our structure. The level of tektin, another suggested IJ protein, did not change as well. Using the predicted molecular weight and fold of the density of the Y-shaped density, we identified FAP276 as the Y-shaped density (details in Materials and methods).

FAP276 supports and mediates the interaction between FAP52 and tubulin (*Figure 2C*). FAP276 itself forms different contacts with tubulin with both of its N- and C-termini, thus it provides strong anchorage for FAP52 to the tubulin lattice (*Figure 2C*). Thus, the IJ complex is made up of two copies of PACRG and FAP20, one copy of FAP52 and FAP276 and one copy of Tether density 1 and 2 per 16 nm (*Figure 2C*). This represents a high stoichiometry compared to other proteins in the axoneme such as CCDC39 and CCDC40, which have only one copy per 96 nm (*Oda et al., 2014*).

PACRG has an alpha solenoid architecture coupled with a long unstructured N-terminal region (*Figure 2—figure supplement 1A*). The alpha solenoid architecture is also contained in the microtubule binding TOG domain, which is present in many microtubule polymerases (*Brouhard et al., 2008*; *Leano and Slep, 2019*). However, the binding orientation of PACRG to the surface of tubulin is completely different from that of the TOG domain (*Leano and Slep, 2019*). FAP20 has a beta jelly roll architecture, which consists of mainly β-sheets with a small α-helix. The C-terminus of FAP20 is located at the outside of the doublet, in agreement with a tomographic study of FAP20 *Chlamydomonas* mutant with a Biotin Carboxyl Carrier Protein tag at the C-terminus (*Yanagisawa et al., 2014*).

PACRG and FAP20 have two potential microtubule-binding sites, one on the surface of the A-tubule and the other on the lateral side of the B-tubule (*Figure 2D,E*). The lateral binding site is unique and has never been observed in previously known microtubule-associated proteins. PACRG binds to the inter-dimer interface of PF B10 in the region of MEIG1 binding loop (*Khan et al., 2019*). In this loop, H136 from PACRG is conserved (*Khan et al., 2019*) and likely interacts with D127 and C129 of α-tubulin from PF B10 (*Figure 2G*). FAP20 is sandwiched by the tubulin dimer from PF B10 and the α-tubulin from PF A1 (*Figure 2D,E*).

The interactions of PACRG and FAP20 with tubulin from PF A1 appear to be electrostatic. The outside surfaces of α- and β-tubulins are highly negatively charged while the corresponding interacting surfaces of PACRG and FAP20 are positively charged (*Figure 2F*).

In addition to the interactions highlighted above, we also observed the possible interaction of the β-tubulin C-terminus from PF A1 with PACRG (*Figure 2E,H*). The C-termini of α- and β-tubulins are a hot spot for post-translational modifications such as polyglutamylation and polyglycylation (*Wloga et al., 2017*). However, due to its flexibility, densities for the α- and β-tubulin C-termini are usually not visible in cryo-EM reconstructions of the microtubules. This is also the case for the outside of the A- and B-tubules in our ex vivo structure. However, in the lumen of the B-tubule, the β-tubulin C-terminus from PF A1 appears to be stabilized by contacts with PACRG and FAP20 (*Figure 2H*). These contacts stabilize the β-tubulin C-terminus forming a helical turn in segment E432-F436, which otherwise would not be present due to its flexibility (*Figure 2H*).

The structure of the β-tubulin C-terminus in PF A1 appears to be the result of the steric proximity with the N-terminus of PACRG. This interaction is important in maintaining the stability of the IJ by preventing steric clashing between the two. It could also be an indication of further post-translational modifications that occur in this region, which could have a potential role in IJ formation and stability.

In our structure, we also observe that the distance between FAP20 and the proximal PACRG is closer compared to the distal PACRG, thus PACRG and FAP20 likely form a heterodimer in the axoneme instead of a continuous filament (*Figure 3A*), except for one missing PACRG unit (*Figure 1—figure supplement 1F,G*). The PACRG and FAP20 binding interface involves complementary surface charges, suggesting a specific and strong interaction (*Figure 3B–D*). The loop N225-I260 of PACRG forms β-sheet-like interactions with strand H33-R36 of FAP20 (*Figure 3B*). In addition, residues Q264 and D258 of PACRG form hydrogen bonds with residue T38 and K20 of FAP20, respectively. This FAP20 binding loop of PACRG is well-conserved among species (*Figure 3E*), but is not present in the PACRG-like protein, a homolog of PACRG that exists in the basal body (*Khan et al., 2019*). FAP20, on the other hand, has a high degree of sequence conservation (*Figure 3—figure supplement 1*).

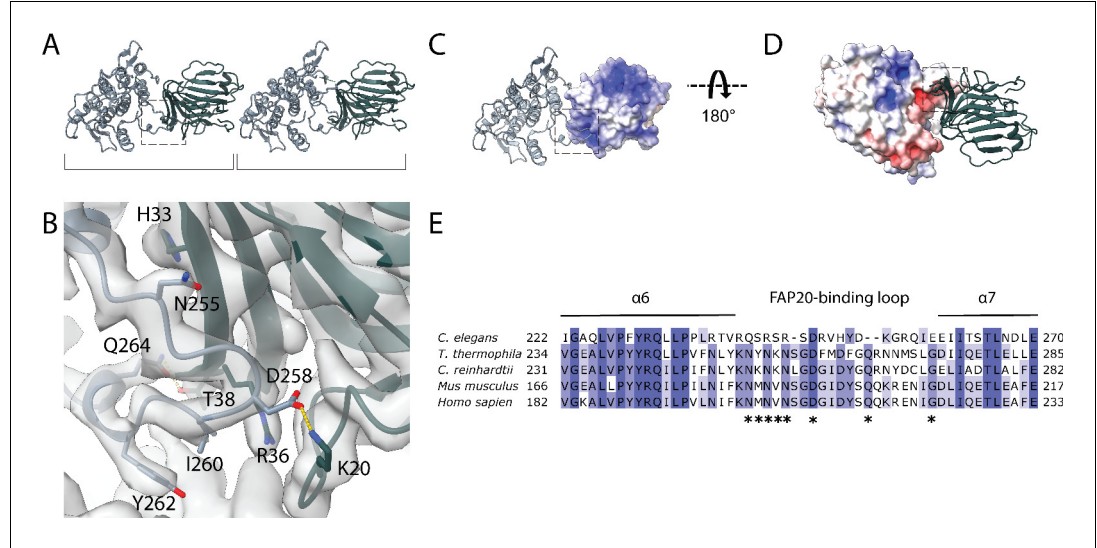

**Figure 3.** Interaction between PACRG and FAP20. (**A**) Consecutive molecules of PACRG and FAP20 in the IJ PF. PACRG and FAP20 form a heterodimer as indicated by brackets. (**B**) Magnified view of the interacting region of PACRG and FAP20. Residues Q264 and D258 of PACRG form hydrogen bonds with residue T38 and K20 of FAP20, respectively. (**C, D**) Electrostatic interactions between PACRG and FAP20 illustrated by their surface charge. The dashed boxes in (**A, C, D**) highlight the interacting loops between PACRG and FAP20 (**B**). (**E**) Multiple sequence alignment of PACRG in the regions of FAP20-binding loop. Asterisks indicate residues that are involved in FAP20 binding.

The online version of this article includes the following figure supplement(s) for figure 3:

**Figure supplement 1.** Multiple sequence alignment of FAP20 shows that it is highly conserved.

The cryo-EM structure of the *Chlamydomonas* PACRG is highly similar to the crystal structure of the human PACRG binding to MEIG1 (PDB: 6NDU) (*Khan et al., 2019*), suggesting a conserved role of PACRG. *Chlamydomonas* PACRG has a long N-terminus that binds on top of PF A13 and into the wedge between PF A1 and A13 (*Figure 2I*, *Figure 2—figure supplement 1A*). This N-terminal region is not conserved in humans or *Tetrahymena* (*Khan et al., 2019*). This could indicate organism-specific adaptations to achieve finely tuned degrees of ciliary stability.

## FAP52 forms an interaction hub and stabilizes α-tubulin's acetylated K40 loop

Next, we investigated the structure of FAP52 (*Figure 2—figure supplement 1E*). FAP52 consists of eight WD40 repeats forming two seven-bladed beta-propellers. The two beta-propellers form a V-shape that docks onto PF B9 and B10. The proximal beta-propeller docks onto the inside of the α- and β-tubulin intra-dimer interface, while the distal beta-propeller is aligned with the next inter-dimer interface toward the plus end (*Figure 4A*).

The distal beta-propeller of FAP52 has a three-point contact with the inner surface of the B-tubule (*Figure 4B*). Two of the FAP52 contacts involve the K40 loop of α-tubulin from PF B9 and B10. The α-K40 acetylation was first discovered in *Chlamydomonas* flagella, which is almost fully acetylated (*LeDizet and Piperno, 1987*). This α-K40 loop has not been fully visualized in reconstituted studies of acetylated tubulins (*Eshun-Wilson et al., 2019*; *Howes et al., 2014*). In our structure, the α-K40 loop is fully structured in this position (*Figure 4C,D* and *Figure 4—figure supplement 1D–G*). The density of the α-K40 loop of PF B10 has a higher SNR than the one from B9. Residue K229 of FAP52 is within favorable distance to form hydrogen bonds with the backbone of D39 and T41 of α-tubulin from PF B10. It is also possible that there is a hydrophobic interaction between L226 and I42 (*Figure 4E*). Residue R225 of FAP52 and D39 of α-tubulin are in an

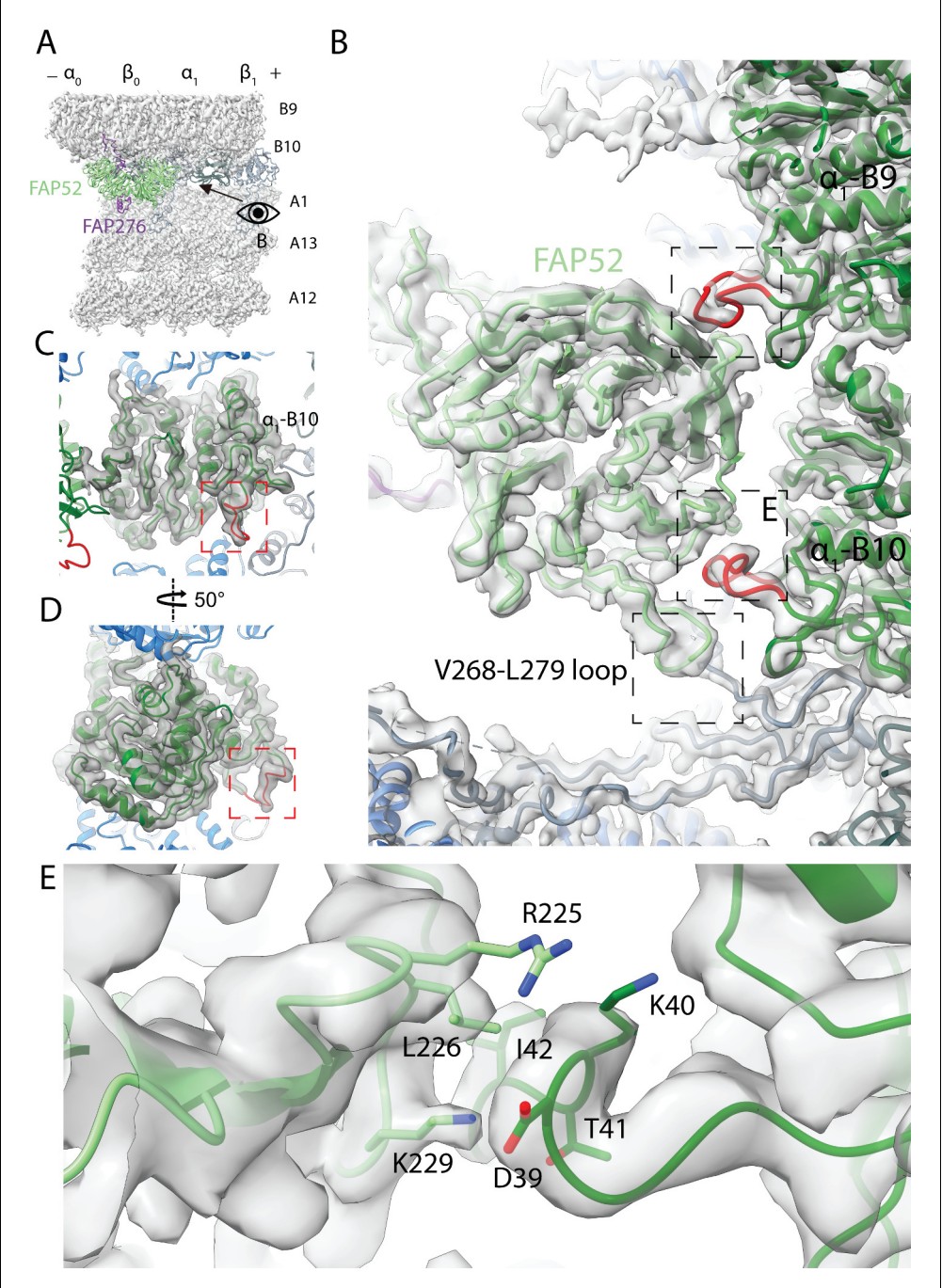

**Figure 4.** Structure of FAP52 and its interaction with tubulins and PACRG. (A) Structure of FAP52 in a top view from the outside of the B-tubule looking down on the A-tubule. Black arrow indicates the direction of view in (B). (B) Three-point contacts of FAP52 with α-tubulins from PF B9 and B10 and PACRG, indicated by the black dashed boxes. The α-K40 loops are colored red. (C, D) The structure of the α-K40 loop from PF B10. Red dashed boxes indicate the α-K40 loop. (E) Interaction of α-K40 loop of PF B10 with FAP52. FAP52's K229 is within favorable distance to form hydrogen bonds with the backbone of D39 and T41. Hydrophobic interactions between L226 and I42 and interactions involving R225 are also possible.

The online version of this article includes the following figure supplement(s) for figure 4:

**Figure supplement 1.** Data related to FAP52.

environment where they could interact; however, the side chain densities for both residues are not well-resolved. At the second tubulin contact point, FAP52 segment G142-P144 is in the close proximity of the α-K40 loop from PF B9 (*Figure 4B*). In the lower region of FAP52, the loop V268-L279 is in close proximity with the N-terminus of PACRG (*Figure 4B*). The density of the aforementioned loop is not present in the FAP52 structure in *Tetrahymena* doublet (*Figure 4—figure supplement 1A,B*). This long loop (V268 to L279) of *Chlamydomonas* FAP52 is, in fact, deleted in other species (*Figure 4—figure supplement 1C*). The interaction of this loop with *Chlamydomonas* PACRG suggests that it is a *Chlamydomonas*-specific feature that contributes to additional stabilization of PACRG and, hence the IJ PF.

We then investigated the α-K40 loops from *Chlamydomonas* and *Tetrahymena* doublets (*Figure 4—figure supplement 1D–G*). When there is no interacting protein, this loop is flexible consistent with previous literature (*Eshun-Wilson et al., 2019*). Despite having low resolution in the B-tubule in *Tetrahymena*, we still observed the α-K40 loop of PF B9 and B10 interacts with FAP52 (*Figure 4—figure supplement 1B*). We were also able to visualize the α-K40 loop in several locations in both *Chlamydomonas* and *Tetrahymena* doublets where there are other proteins interacting with it (*Figure 4—figure supplement 1F,G*). The conformation of the α-K40 loop appeared to be different depending on its corresponding partner. This suggests that the α-K40 loop might have a role in MIP recognition and binding. Given the numerous interactions of FAP52 with all the proteins in the IJ, FAP52 is likely to function as an interaction hub, which could play an important role during IJ assembly.

## FAP106 is the Tether loop, consisting of Tether densities 1 and 2

We were able to trace the Tether density in the 16 nm averaged map. Tether density 1 is connected to Tether density 2 (*Figure 5A–D*), forming a Tether loop, through which the A- and B-tubules are connected. The loop connecting the Tether density 1 binds on top of PF A12 and then into the outside wedge between A12 and A13 before connecting with Tether density 2. The entire Tether loop is a single polypeptide, conserved between *Tetrahymena* and *Chlamydomona*s (*Figure 5A–B*). Part of this Tether loop resembles Tau binding to the microtubule (*Kellogg et al., 2018*). There is a small helical region in this loop that binds to α-tubulin of PF A12 (*Figure 5—figure supplement 1C,D*).

To identify the protein that makes up the Tether loop, we utilized mass spectrometry, bioinformatics and modeling (details in Materials and methods, *Figure 5—figure supplement 1A*). This allowed us to identify the Tether loop as FAP106. We were able to model segments Q2-P13, P20-A148 and W189-I226 where the density had sufficient signal (*Figure 5C,D* and *Figure 5—figure supplement 1B*). Helix H3 and H4 of FAP106 insert into the interdimer interfaces between PF B9 and B10 forming the anchor point to the B-tubule (*Figure 5E*) while helix H1 and H2 bind to β-tubulin of PF A13 and α-tubulin of PF A12 (*Figure 5D*). FAP106 is a homolog of ENKURIN (ENKUR), a conserved protein in sperms of many species (*Sutton et al., 2004*; *Jungnickel et al., 2018*). Enkur knockout mice have abnormal sperm motility with asymmetric flagellar waveform and therefore low fertility rate (*Jungnickel et al., 2018*). In addition, mutations in ENKUR is linked to *situs inversus* in human and mouse (*Sigg et al., 2017*; *Stauber et al., 2017*). However, the IQ motif of Enkurin that binds Calmodulin is not conserved in *Chlamydomonas* (*Figure 5—figure supplement 1E*).

In *Tetrahymena*, Tether density 3 connects the distal lobe of FAP52 and binds across the wedge between PF A13 and A1. (*Figure 5F*). Upon superimposing the *Chlamydomonas* PACRG structure onto the *Tetrahymena* IJ area, the N-terminus of PACRG will have a steric clash with Tether density 3 (*Figure 5G*). This explains the shorter N-terminus of *Tetrahymena* PACRG relative to the *Chlamydomonas* PACRG. Tether density 3 might interact with and perform the same function as the N-terminus of PACRG in *Chlamydomonas*.

## FAP126, a FLTOP homolog, interacts with the tether loop, FAP106

Using the same approach as FAP106, we were also able to identify a density that lies on top of PF A13 and goes into the wedge between PF A12 and A13 (*Figure 6A*, turquoise and *Figure 6—figure supplement 1B*) as FAP126. FAP126 does not have a homolog in *Tetrahymena* and is not present in the *Tetrahymena* map.

FAP126 is a homolog of the human FLTOP protein, which is shown to be important for basal body docking and positioning in mono- and multi-ciliated cells (*Gegg et al., 2014*). Multiple

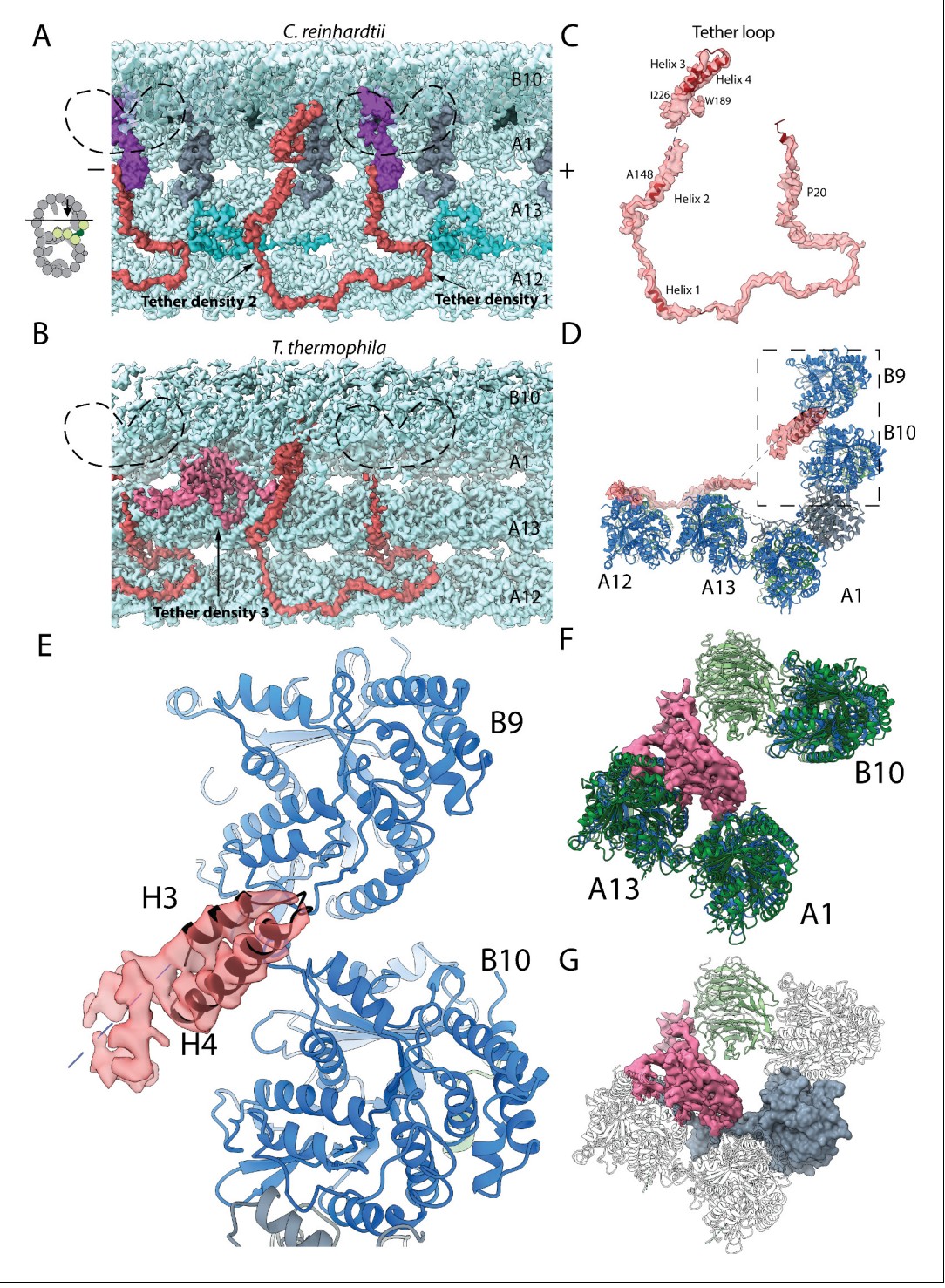

**Figure 5.** Structure of the Tether densities. (**A–B**) At higher resolution, Tether densities 1 and 2 appear to be a single polypeptide chain in both *Chlamydomonas* (**A**) and *Tetrahymena* (**B**). Color scheme: tubulin: transparency gray; Tether densities 1 and 2 (Tether loop/FAP106): red; FAP276: purple; PACRG: light gray; Densities between PF A13 and A12 (turquoise). The dashed regions indicate the location of FAP52, which has been digitally removed to show the Tether densities underneath. (**C**) Model of FAP106 fitted inside the segmented Tether loop from *Chlamydomonas*. (**D**) Model of FAP106 tethering the B-tubule and A-tubule. Dashed box indicates view in (**E**). (**E**) Helix H3 and H4 of FAP106 insert into the gap formed by four tubulin dimers of PF B9 and B10. (**F**) Structure of Tether density 3 from *Tetrahymena*, which binds on top of the wedge between PF A13 and A1. (**G**) Overlay of the

*Figure 5 continued on next page*

*Figure 5 continued*

PACRG from *Chlamydomonas* onto the structure of *Tetrahymena* shows a hypothetical steric clash of a long *Tetrahymena* PACRG N-terminus with Tether density 3.

The online version of this article includes the following figure supplement(s) for figure 5:

**Figure supplement 1.** Data related to the Tether densities.

alignment sequence alignment of FAP126 shows that the *Chlamydomonas* FAP126 lacks the proline-rich regions of other species (*Figure 6—figure supplement 1A*).

FAP126 appears to interact with FAP106 (*Figure 6B*). Segment F75-Q77 of FAP126 is in proximity to segment Q126-Q129 of FAP106. T76 and Q77 of FAP126 form hydrogen bonds with Q126 and Q129 of FAP106, respectively (*Figure 6B*). Therefore, FAP126 might play a role in recruiting FAP106 or vice versa. Almost half of FAP126 density runs along the wedge between PF A12 and A13, close to the tubulin lateral interface with complementary surface charge (*Figure 6C,D*). FAP126 might act as a low curvature inducer or sensor from the outside similar to Rib43a from the inside since the curvature of A12 and A13 is significantly lower compared to 13-PF singlet (*Figure 6E,F*) (*Ichikawa et al., 2019*).

To support whether FAP126 interacts with FAP106, we analyzed the normalized RNA expression of FAP126 with FAP106 (ENKUR) and FAP52 from different human tissues (*Figure 6G,H* and *Figure 6—figure supplement 1D–F*). FAP126 showed high correlation with both FAP106 (ENKUR) and FAP52(r = 0.89, p-value=<0.0001, r = 0.94, p-value=<0.0001, respectively) (*Figure 6G,H*). This indicates that FAP126 might be functionally related to other members of the IJ complex such as FAP106 and FAP52, which further supports the identity of these proteins.

## Discussion

In this study, we describe the molecular details of the IJ complex using a combination of mass spectrometry and cryo-EM. The IJ complex in *Chlamydomonas* is made up of PACRG, FAP20, FAP52, FAP276, FAP106 (Tether loop) and associated proteins such as FAP126 and FAP45 (*Figure 7A*) and *Video 1*). We identified two new members of the IJ, FAP106 and FAP276. FAP276, a *Chlamydomonas*-specific protein, anchors and mediates FAP52's binding onto tubulins from PF B9 and B10. FAP106 tethers the B-tubule to the A-tubule, through its interactions with the PF A12 and A13, FAP52 and FAP276, while the IJ PF, composed of PACRG and FAP20, closes the IJ gap. For the doublet to withstand the mechanical strain during ciliary beating, it needs proper structural supports. Tektin, a coiled-coil protein, was also proposed to be another component of the IJ complex in *Chlamydomonas* by biochemical experiments (*Yanagisawa et al., 2014*). However, no filamentous density corresponding to tektin was found at the IJ PF in our *Chlamydomonas* map. It suggests that tektin in *Chlamydomonas* might not be located inside the doublet and is washed out from the salt treatment.

Reconstituted doublet microtubules (*Schmidt-Cernohorska et al., 2019*) indicate that the B-tubule cannot be closed and is extremely flexible without the IJ PF. Therefore, the IJ PF is necessary to dock the B-tubule onto the A-tubule. In our *Tetrahymena* doublet, in which most of the IJ PF was washed away, even with the presence of FAP52 and FAP106, the doublet is still flexible which can be seen by the lower resolution of the B-tubule compared to the A-tubule (*Figure 1—figure supplement 1D*). In addition, the B-tubule can be subjected to depolymerization when the IJ PF is not fully formed (*Owa et al., 2019*). Therefore, the IJ PF serves as an anchor, which might protect the B-tubule from depolymerization by shielding the lateral side of PF B10 (*Figure 7B*). Because of the complexity of interactions and the diverse protein composition of the IJ complex, it is reasonable to assume that the IJ is assembled after the outer junction nucleates and expands toward the IJ. The IJ complex might be assembled or co-assembled at the same time as PF B10 for the closure of the B-tubule (*Figure 7B*).

During doublet assembly, the unorderly binding of PACRG and FAP20 to any of the PFs in the B-tubule lateral interfaces would lead to an incomplete B-tubule (*Khan et al., 2019*). To ensure a

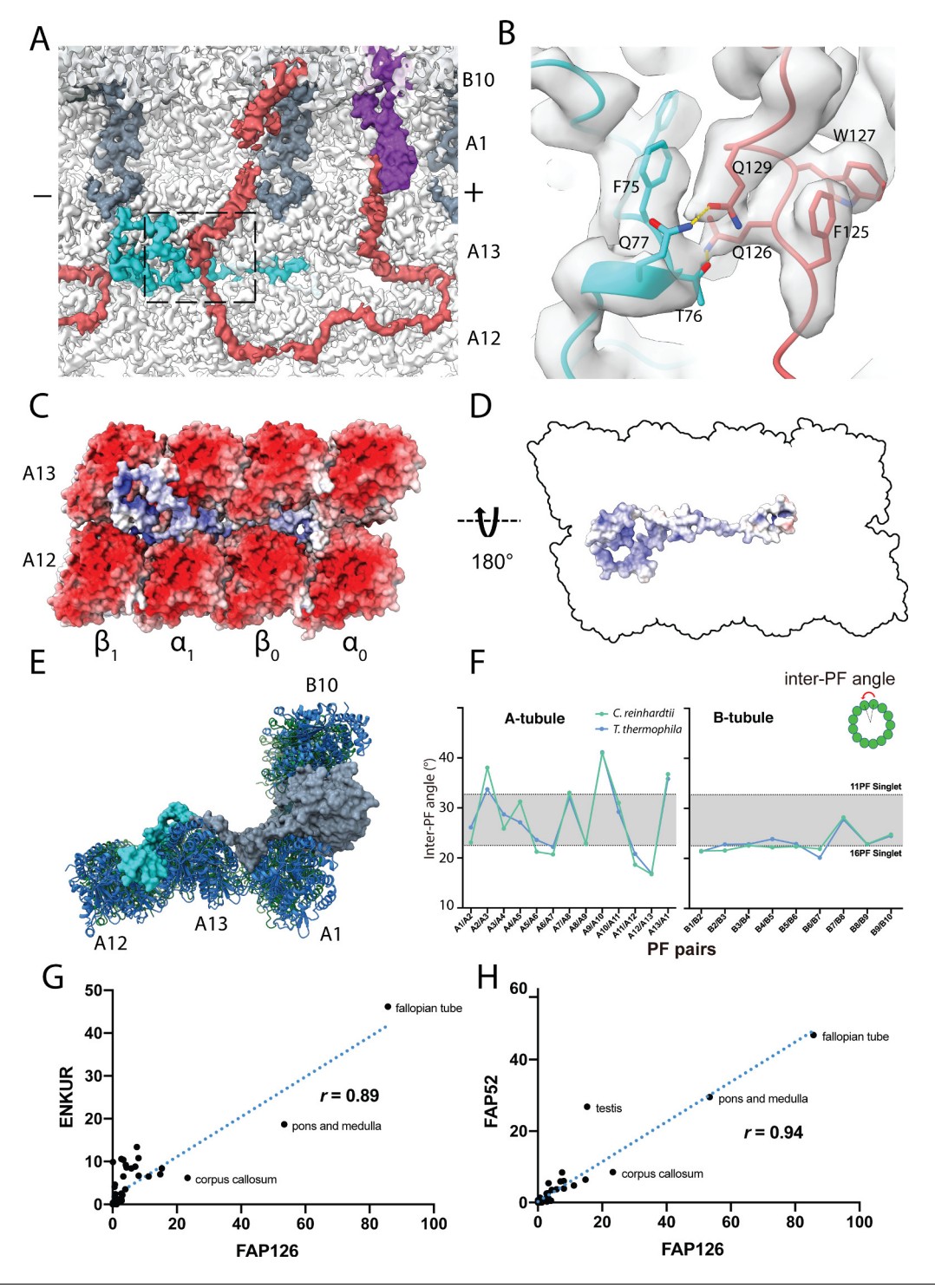

**Figure 6.** Structure of FAP126 and its interaction. (**A**) View from the top of the PF A12 and A13 showing the density of FAP126 (dark turquoise). Dashed box indicates view in (**B**). (**B**) Closeup view of the interactions of FAP126 with the Tether loop, FAP106. T76 and Q77 of FAP126 form hydrogen bonds with Q126 and Q129 of FAP106, respectively. (**C**) Complimentary electrostatic surface charges of tubulins and FAP126. (**D**) Electrostatic charge of FAP126 on the tubulin interacting surface. (**E**) The N-terminus of PACRG and the hook density go into the wedges between PF A12 and A13, and PF A13 and A1, respectively. This likely contributes to the curvature of this region. (**F**) Inter-PF angles of the A- and B-tubules from *Chlamydomonas* and *Tetrahymena* (***Ichikawa et al., 2019***) showing very similar angle distributions. (**G** and **H**) Correlation graphs of consensus normalized expression

*Figure 6 continued on next page*

*Figure 6 continued*
levels for two selected pairs of genes (ENKURIN(FAP106)/FAP126 and FAP52/FAP126). Tissues showing high levels of expression of one or both genes are labeled. Correlation coefficients (r) are indicated. By excluding testis, corpus callosum, pons and medulla and fallopian tube, the correlation coefficients between ENKUR and FAP126 and between FAP52 and FAP126 are 0.66 and 0.84, respectively.
The online version of this article includes the following figure supplement(s) for figure 6:

**Figure supplement 1.** Data related to FAP126.

successful IJ assembly, chaperones might be needed for the transport of PACRG and FAP20. PACRG forms a complex with MEIG1 (*Khan et al., 2019*). Even though MEIG1 is not present in lower eukaryotes and the MEIG1 binding loop is not conserved between *Chlamydomonas* and humans, a chaperone similar to MEIG1 can function to target PACRG to the lateral interface of the PF B10. There is a possibility that PACRG and FAP20 form a heterodimer before their transport and assembly into the cilia. FAP20 shows a similar fold and mode of binding to a class of proteins called carbohydrate-binding modules. Carbohydrate-binding modules form a complex with carbohydrate-active enzymes and are known to have a substrate targeting and enzyme-concentrating function (*Hervé et al., 2010*). This supports the role of FAP20 as an assembly chaperone in a FAP20-PACRG complex. Furthermore, both studies from *Yanagisawa et al. (2014)* and *Dymek et al. (2019)* show reduced endogenous PACRG in *Chlamydomonas* FAP20 knockout mutant. In the latter study, it was shown that the assembly of exogenous PACRG was less efficient in the FAP20 knockout compared to conditions where FAP20 was intact. This implies that PACRG assembly might indeed depend on FAP20 (*Yanagisawa et al., 2014*). However, since the expression patterns of PACRG and FAP20 have surprisingly low correlation compared to the rest of the IJ proteins (*Figure 6—figure supplement 1F*) and homolog of FAP20 exists in non-ciliated organisms such as *Arabidopsis*, FAP20 might have an additional function outside the cilia.

Furthermore, our atomic models could explain the severe motility phenotypes observed in PACRG and FAP20 mutants compared to FAP52 mutant. Mutants in either PACRG or FAP20 might affect the stability of the DRC, which can severely affect the regulation of ciliary beating. This is supported by the fact that FAP20 mutant is prone to splaying of the cilia (*Yanagisawa et al., 2014*). Our results could also explain how the double knockout of FAP20 along with FAP45 or FAP52 can affect B-tubule stability at the IJ (*Owa et al., 2019*). In such conditions, both the IJ PF and the FAP52 or FAP52-mediated anchorage between the A- and B-tubules will be completely lost.

By comparing *Chlamydomonas* and *Tetrahymena*, we show that the conserved IJ components are PACRG, FAP20, FAP45, FAP52 and FAP106. There are also species-specific proteins such as FAP276 and FAP126 in *Chlamydomonas* and Tether density 3 in *Tetrahymena*. Tether density 3 clashes with a superimposed *Chlamydomonas* FAP276 structure, suggesting that it takes over its role in mediating the interactions between FAP52 and tubulin in *Tetrahymena*. This suggests that there is a common framework for the IJ complex in all species. Species-specific proteins may then fine-tune this framework according to the survival needs of the organisms.

In this study, we revealed that FAP106/ENKUR, an important protein for sperm motility, is a MIP and an IJ protein. Knockout of ENKUR leads to the asymmetric waveform of sperm flagella while mutations in ENKUR disturb the left-right symmetry axes in vertebrates. It is shown that ENKUR knockout shows a loss of $Ca^{++}$ responsiveness while wild-type sperm shows highly curved flagella (*Jungnickel et al., 2018*). The IQ domain responsible for $Ca^{++}$ binding of ENKUR is not conserved in the *Chlamydomonas* sequence, although an alternative means of $Ca^{++}$ binding or inducing a $Ca^{++}$-mediated response is still possible.

We also identified FAP126, an IJ-associated MIP. The homolog of FAP126 in human and mouse, the FLTOP protein, exists in the cilia and basal bodies and is thought to function in the positioning of the basal body (*Gegg et al., 2014*). In Flattop knockout mice, cilia formation in the lung is significantly affected. In the inner ear, Flattop interacts with a protein called Dlg3 in the process of basal body positioning to the actin skeleton in the inner ear. Therefore, it is possible that FAP126 might perform both functions (i) as a MIP that stabilizes the basal body in the same fashion as shown here

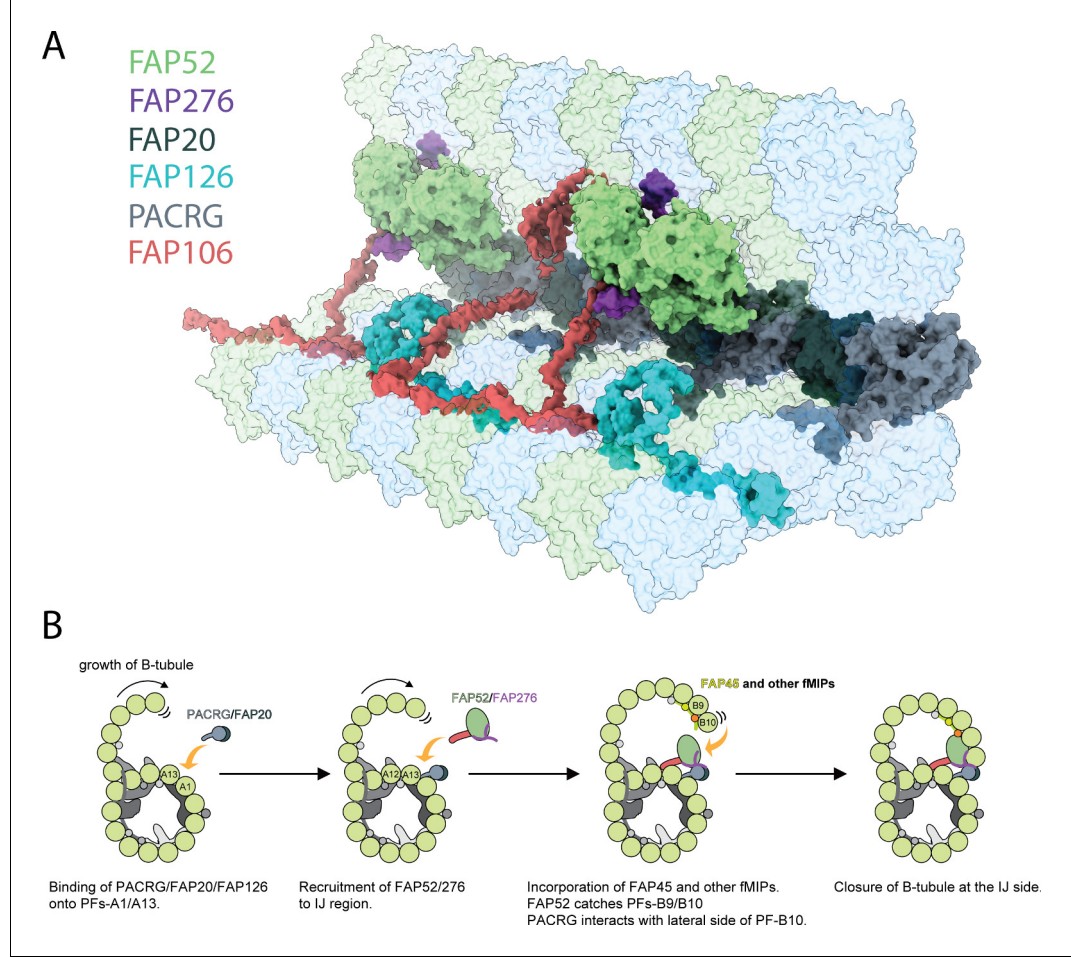

**Figure 7.** Inner junction structure and proposed model of IJ formation. (**A**) Model of the IJ complex including PACRG, FAP20, FAP52, FAP126, FAP276 and FAP106. FAP45 is not depicted here. Tubulin is depicted as transparent. (**B**) The B-tubule starts growing laterally from the outer junction side as shown in *Schmidt-Cernohorska et al. (2019)*. PACRG and FAP20 form a heterodimer, which binds onto the outside surface of PF A1. Following this, multiple alternative hypotheses are possible. One hypothesis is that FAP52, FAP276 and FAP106 would bind onto PF A12 and A13. FAP45 and other fMIP proteins would then be incorporated inside the B-tubule, which fixes the proper curvature so that PF B9 and B10 can interact with other IJ proteins. FAP52 binds both PF B9 and B10 through their K40 loops and finally, PACRG and FAP20 interact with the lateral side of PF B10 allowing for B-tubule closure.

in the cilia and (ii) in basal body positioning and planar cell polarity (*Gegg et al., 2014*). The high correlation of FAP126 and FAP106 co-expression, and also with FAP52 in different human tissue suggests they might function similarly or co-operatively in cilia assembly. *Tetrahymena*, which lacks FAP126, probably implements an alternative mechanism to substitute for FAP126 functions.

Our study demonstrates that all the protein components associated with the IJ complex such as PACRG, FAP20, FAP52, FAP126 and FAP106 in the IJ complex is of high importance for the assembly and proper motility of the cilia. Multiple studies have indeed showed the implication of such proteins in human disease (*Lorenzetti et al., 2004*; *Ta-Shma et al., 2015*; *Stauber et al., 2017*). Remarkably, these proteins are MIPs existing inside the doublet except for PACRG and FAP20. In addition, the structures of the IJ components such as PACRG and FAP126 also highlight the unique roles of the MIPs in curvature induction or sensing as shown previously with Rib43a (*Ichikawa et al., 2019*). FAP126 binds tightly to the wedge PF A12 and A13 while the N-terminus of *Chlamydomonas*

PACRG penetrates the wedge between PF A13 and A1 and forces the PF pairs into a high-curvature conformation (*Figure 6E*). From our curvature analysis of the doublet (*Figure 6F*), this region of PF A12-A1 contains extreme high and low curvatures compared to the 13-PF singlet. Alternatively, the curvature might be enforced by MIPs inside the A-tubule. This inter-PF curvature could help to facilitate the specific binding and anchoring of FAP126 and PACRG to the correct position. It has been shown that doublecortin can sense the curvature of the 13-PF microtubule (*Bechstedt and Brouhard, 2012*). The *Tetrahymena*-specific Tether density 3 might act as a high curvature inducer/sensor or an IJ complex stabilizer.

Post-translational modifications in tubulin are known to be important for the activity of the cilia. There have been many studies about the effect of acetylation on the properties of microtubule such as stability (*Portran et al., 2017*; *Xu et al., 2017*). In 3T3 cells, the K40 acetyltransferase, $\alpha$TAT1 promotes rapid ciliogenesis (*Friedmann et al., 2012*). The absence of acetylating enzymes has indeed been shown to affect sperm motility in mice (*Kalebic et al., 2013*) while SIRT2 deacetylation decreases axonemal motility in vitro (*Alper et al., 2014*). A recent cryo-EM study of reconstituted acetylated microtubules showed, using molecular dynamics, that the acetylated $\alpha$-K40 loop has less conformational flexibility, but a full $\alpha$-K40 loop in the cryo-EM map has not been visualized due to its flexibility. In this work, we show that the acetylated K40 loop binds to FAP52 and forms a fully structured loop. This loop remains flexible and unstructured when there is no protein interacting with it. This suggests that the $\alpha$-K40 loop has a role in protein recruitment and interactions, especially, MIPs. We hypothesize that the acetylation disrupts the formation of an intra-molecular salt bridge between K40 and D39, which affects the loop's sampling conformations and allows D39 to take part in atomic interactions with other proteins. This, in turn, improves the stability of the doublet and therefore, correlates with axonemal motility. In neurons, microtubules are also highly acetylated and are known to be stable. Our hypothesis suggests that in neuron microtubules, there might exist MIPs with a similar stabilizing effect as in the doublet. Previous studies on olfactory neurons demonstrate that there are densities of proteins inside the microtubule, suggesting the existence of MIPs inside cytoplasmic microtubules (*Burton, 1984*).

Another interesting insight from our study is the structured C-terminus of β-tubulin. The C-termini of tubulin in the doublet normally have polyglycylation and polyglutamylation, in particular, the B-tubule (*Lechtreck and Geimer, 2000*). In reconstituted microtubules and other places in the doublet, the C-termini are highly flexible and cannot be visualized. However, we observed the C-terminus of β-tubulin in PF A1 which appears to interact with PACRG and FAP20. In addition, the position of FAP126 and FAP106 binding on top of tubulin molecules also suggest they are interacting with the C-termini of tubulins. In vitro study shows that the C-tails of tubulins must be suppressed for the outer junction to be formed (*Schmidt-Cernohorska et al., 2019*). This suggests that the C-terminus might have a role in the assembly and or stability of the doublet. Defects in tubulin polyglutamylase enzyme have indeed led to partially formed B-tubules (*Pathak et al., 2007*). This could indicate a role for polyglutamylation in the interaction and recruitment at the IJ PF, specifically PACRG and FAP20. Lack of polyglutamylation can lead to an easily detachable PACRG and FAP20 and hence the partial assembly of the B-tubule. Finally, it is possible that MIPs can act as readers of tubulin post-translational modifications for their orderly recruitment and assembly.

## Materials and methods

### Preparation of doublet samples

WT *Chlamydomonas* cells (cc124) were obtained from *Chlamydomonas* source center and cultured either on Tris-acetatephosphate (TAP) media with shaking or stirring with 12 hr light-12 hr dark cycle. For flagella purification, *Chlamydomonas* cells were cultured in 1.5 L of liquid TAP media with stirring until OD600 reached around 0.5–0.6 and harvested by low-speed centrifugation (700 g for 7 min at 4℃). *Chlamydomonas* flagella were purified by dibucaine method (*Witman, 1986*), resuspended in HMDEKP buffer (30 mM HEPES, pH 7.4, 5 mM MgSO4, 1 mM DTT, 0.5 mMc, 25 mM Potassium Acetate, 0.5% polyethylene glycol, MW 20,000) containing 10 μM paclitaxel, 1 mM PMSF, 10 μg/ml aprotinin and 5 μg/ml leupeptin. Paclitaxel was added to the buffer since *Chlamydomonas* doublets were more vulnerable to high-salt extraction compared with *Tetrahymena* doublets (data not shown). Isolated flagella were demembraned by incubating with HMDEKP buffer containing final

1.5% NP40 for 30 min on ice. After NP40 treatment, *Chlamydomonas* doublets were incubated with final 1 mM ADP for 10 min at room temperature to activate dynein and then incubated with 0.1 mM ATP for 10 min at room temperature to induce doublet sliding. Since the *Chlamydomonas* doublets were harder to split compared to *Tetrahymena* doublet, sonication was done before ADP/ATP treatment. After this, *Chlamydomonas* doublets were incubated twice with HMDEKP buffer containing 0.6 M NaCl for 30 min on ice, spinned down (16,000 g and 10 min), and resuspended. *Chlamydomonas* doublets were not dialyzed against low-salt buffer since it was difficult to remove radial spokes.

Tetrahymena doublets were isolated according to our previous work (*Ichikawa et al., 2017*; *Ichikawa et al., 2019*).

## Cryo-electron microscopy

3.5 μl of sample of sonicated doublets (~4 mg/ml) was applied to a glow-discharged holey carbon grid (Quantifoil R2/2), blotted and plunged into liquid ethane using Vitrobot Mark IV (Thermo Fisher Scientific) at 25°C and 100% humidity with a blot force 3 or four and a blot time of 5 s.

9528 movies were obtained on a Titan Krios (Thermo Fisher Scientific) equipped with Falcon II camera at 59,000 nominal magnification. The pixel size was 1.375 Å/pixel. Dataset for *Tetrahymena* was described in *Ichikawa et al. (2019)*. *Chlamydomonas* dataset was collected with a dose of 28–45 electron/Å2 with seven frames. The defocus range was set to between −1.2 and −3.8 μm.

The *Chlamydomonas* doublet structures were performed according to *Ichikawa et al. (2019)*. In short, movies were motion corrected using MotionCor2 (*Zheng et al., 2017*). The contrast transfer function were estimated Gctf (*Zhang, 2016*). The doublets were picked using e2helixboxer (*Tang et al., 2007*).

270,713 and 122,997 particles were used for the reconstruction of 16 nm and 48 nm repeating unit of *Chlamydomonas*. 279,850 particles were used for the 16 nm reconstruction of *Tetrahymena*. The final Gold Standard FSC resolutions of the 16 nm and 48 nm reconstruction for *Chlamydomonas* after contrast transfer function refinement and polishing using 0.143 FSC criterion in Relion3 (*Zivanov et al., 2018*) are 4.5 and 3.8 Å, respectively. Using focus refinement of the IJ of the 16 nm reconstruction for *Chlamydomonas*, the resolution reaches 3.6 Å. The resolution for the 16 nm reconstruction of *Tetrahymena* was 3.6 Å. Focus refinement of the IJ of *Tetrahymena* did not improve the resolution of the IJ due to the flexibility of this region. The maps were local sharpened (*Ichikawa et al., 2019*). Local resolution estimation was performed using MonoRes (*Vilas et al., 2018*).

## Modeling

### *Chlamydomonas reinhardtii* α-β-tubulin

A homology model of *C. reinhardtii* α-β-tubulin (Uniprot sequence α: P09204, β: P04690) was constructed in Modeller v9.19 (*Webb and Sali, 2014*) using the Taxol structure (PDB ID: 5SYF) as template. The model was refined using real-space refinement (*Afonine et al., 2018*) and validated using comprehensive validation for cryo-EM in Phenix v1.16 (*Adams et al., 2010*).

### PACRG and FAP20

A partial homology model of *C. reinhardtii* PACRG (B1B601) was constructed using the crystal structure of the human homolog (Q96M98-1) as template (*Khan et al., 2019*). The model was completed by building segments N2-D148 and Y249-L270 de novo in density using Coot v0.8.9.1 (*Emsley et al., 2010*). The density for segment M89-K101 is missing, likely due to flexibility and or the 16 nm averaging of the cryo-EM data. The missing segment was worked out based on the fact that i) all the surrounding densities have been assigned to the α-β-tubulin heterodimers, ii) there is a 101 long segment of the PACRG N-terminus that is still unassigned to density and iii) the density signature downstream and upstream of the gap matches the sequence identity of the PACRG N-terminus as shown in *Figure 2—figure supplement 1B* (right side). *C. reinhardtii* FAP20 (A8IU92) was completely built de novo in density. Both models were refined and validated as described for α-β-tubulin.

## FAP52

The density was traced in Coot v0.8.9.1 (*Emsley et al., 2010*) based on the topology of a seven-bladed beta propeller (PDB ID: 2YMU), which agrees with the I-TASSER (*Yang et al., 2015*) tertiary structure prediction of FAP52. The Uniprot sequence for *C. reinhardtii* FAP52 (A8ILK1, 615 amino acids long) is missing loop V264-K277. This was worked out based on the sequence-density disagreement following Q263. Upon further investigation, a longer sequence in the Uniprot database (633 long) was identified by a blast search (Uniprot ID: A0A2K3D260), which agrees with our electron density and both the Phytozome transcript (Cre12.g489750.t1.2) and the *C. reinhardtii* FAP52 sequence reported in *Owa et al. (2019)*. Overall, the electron density for FAP52 had less resolution and lower SNR than surrounding proteins. This is likely due to the 16 nm averaging of the cryo-EM data and the heterogeneity of FAP52 due to the 48 nm periodic association with FAP45. The bulky residues of FAP52 were used as anchors to maintain the correct registry in lower resolution areas. The model could be overfit in segment D341-P627 where the density signal is significantly lower. Tracing the backbone of this segment was possible due to the predictability of the seven-bladed beta propeller topology. Furthermore, 7 out of 10 Tryptophan residues of the FAP52 sequence are located within this segment, which served as anchors and allowed us to maintain registry and overfit some side chains where the density was missing. The final model was refined and validated as described above.

## FAP276

After resolving all the proteins in the its environment, the density for the Y-shaped protein was segmented and traced to around 80 amino acids of ~9 kDa in mass. Candidates of approximately this size from the wild-type mass spectrometry data were compared to the FAP52 knockout data and reduced to only FAP276, which was completely missing in the latter (*Table 1*). The secondary structure prediction (*Drozdetskiy et al., 2015*) as well as the sequence of FAP276 (Phytozome transcript: Cre04.g216250) agree with the density signature of the Y-shaped region (*Figure 2—figure supplement 1G,H*). The model was traced, refined and validated as described above.

## FAP106

The trace of the Tether loop was estimated to be ~220–240 amino acids due to missing and likely flexible segments of this protein. To identify the protein that makes up the Tether loop, this protein needs to satisfy the following criteria: (i) has a high stoichiometry (1 per 16 nm of the doublet), (ii) has a minimum molecular weight of ~25 kDa (based on the sequence trace) and (iii) conserved in both *Chlamydomonas* and *Tetrahymena*.

We calculated the stoichiometry of proteins in the doublet after salt extraction by normalizing the averaged quantitative spectral count of each protein by their molecular weight. The triplicate mass spectrometry data comes from *Dai et al. (2019)*. The top 35 proteins by copy numbers are shown in *Table 2*. In our calculation, some radial spoke and central pair proteins displayed high stoichiometry such as RSP9 and PF16. Remarkably, all the IJ proteins are in the top 35 (PACRG, FAP52, FAP20, FAP45 ranked 4, 9, 10 and 35, respectively) as supported by our structure. This validates the quality of the stoichiometry calculation. Although FAP276 should have the same stoichiometry as FAP52, it does not appear in high stochiometric numbers. This can be explained that by the small size of FAP276, which is not well detected in mass spectrometry.

Among the proteins that have high stoichiometry, the following proteins satisfy the three criteria above: FAP115, FAP106, FAP252, FAP161, FAP77 and FAP71. However, the homologs of FAP115 and FAP161 in *Tetrahymena* are too big. Our analysis of the secondary structure prediction places FAP106 at the top of the list of candidates for the Tether loop (*Figure 5C*).

The identity of FAP106 was further confirmed by a blast search against the complete *C. reinhardtii* proteome using a regular expression pattern that matches the density signature around residue W127 ([FHY]xWxxKxx[FHY]) (*Figure 5—figure supplement 1A*). The search was restricted for sequences of lengths between 220 and 280 based on the sequence trace. This returned two matches: FAP106 (uniport ID: A8IVJ1) and a transcription factor (uniport ID: A8I9A1). Furthermore, the sequence secondary structure prediction of FAP106 had high confidence in four α-helices and a long-disordered segment, which agrees with the density topology of this protein. As before, the sequence of FAP106 had side chain agreement with the density throughout the entire sequence.

**Table 2.** Normalized spectral count of proteins detected by mass spectrometry.

| Name | Molecular Weight in kDa (MW) | Average quantitative Value (AQV) | Rough Stoichiometric Peptide Abundance (RSPA)* | *T. thermophila* Homologs** | Human homologs*** | Localization in *C. reinhardtii* |
|------|------|------|------|------|------|------|
| TUA1 | 50 | 1077.97 | 215.59 | TBA_TETTH | TUBA1C | Doublet |
| TUB1 | 50 | 625.46 | 125.09 | TBB_TETTH | TUBB4B | Doublet |
| RIB72 | 72 | 116.72 | 16.21 | TTHERM_00143690 | EFHC1 | MIP |
| PACRG | 25 | 38.39 | 15.35 | TTHERM_00446290 | PACRG | IJ |
| PF16 | 50 | 74.09 | 14.82 | TTHERM_000157929 | SPAG6 | Central Pair |
| RSP9 | 30 | 41.79 | 13.93 | TTHERM_00430020 | RSPH9 | Radial Spoke |
| FAP86 | 30 | 36.11 | 12.04 | - | - | Doublet |
| FAP1 | 22 | 26.46 | 12.03 | - | - | Doublet |
| FAP52 | 66 | 79.12 | 11.99 | TTHERM_01094880 | CFAP52 | MIP |
| FAP20 | 22 | 26.08 | 11.86 | TTHERM_00418580 | CFAP20 | IJ |
| FAP126 | 15 | 16.89 | 11.26 | - | CFAP126 | MIP |
| RSP1 | 88 | 98.73 | 11.22 | TTHERM_00047490 | RSPH1 | Radial Spoke |
| FAP115 | 27 | 29.98 | 11.10 | TTHERM_00193760 | - | Doublet |
| FAP106 | 27 | 29.81 | 11.04 | TTHERM_00137550 | ENKUR | IJ? |
| Tektin | 53 | 57.61 | 10.87 | - | TEKT5 | IJ? |
| RSP3 | 57 | 60.55 | 10.62 | TTHERM_00566810 | RSPH3 | Radial Spoke |
| FAP252 | 39 | 39.97 | 10.25 | TTHERM_00899430 | CETN3 | Axonemal |
| RSP2 | 77 | 77.95 | 10.12 | - | CALM2 | Radial Spoke |
| FAP161 | 43 | 43.50 | 10.12 | TTHERM_00155380 | CFAP161 | Axonemal |
| IDA4 | 29 | 27.29 | 9.41 | TTHERM_00841210 | DNALI1 | Dynein |
| FAP107 | 26 | 23.66 | 9.10 | - | FLG2 | Axonemal |
| DHC2 | 457 | 414.12 | 9.06 | TTHERM_01027670 | DNAH1 | Dynein |
| FAP12 | 54 | 48.85 | 9.05 | - | DAGLB | Cytoplasmic |
| RSP7 | 34 | 30.62 | 9.01 | TTHERM_00194419 | CALML5 | Radial Spoke |
| RSP5 | 56 | 49.32 | 8.81 | - | - | Radial Spoke |
| FAP230 | 45 | 39.59 | 8.80 | - | - | Axonemal |
| FAP77 | 29 | 23.88 | 8.24 | TTHERM_00974270 | CFAP77 | Axonemal |
| FAP55 | 111 | 90.47 | 8.15 | - | MYH14 | Axonemal |
| FAP90 | 28 | 22.35 | 7.98 | - | WBP11 | Axonemal |
| RSP10 | 24 | 19.10 | 7.96 | TTHERM_00378600 | RSPH1 | Radial Spoke |
| FAP71 | 32 | 24.86 | 7.77 | TTHERM_00077710 | EWSR1 | Axonemal |
| EEF1 | 51 | 39.04 | 7.66 | TTHERM_00655820 | Multiple | Axonemal |
| FAP182 | 49 | 36.69 | 7.49 | TTHERM_01049330 | C9orf116 | Axonemal |
| Rib43a | 43 | 32.06 | 7.46 | TTHERM_00624660 TTHERM_00641119 | RIBC2 | MIP |
| FAP45 | 59 | 43.20 | 7.32 | TTHERM_001164064 | CFAP45 | MIP |

*RSPA was calculated by (AQV)/(MW)*10.

** *T. thermophila* homologs were BLAST searched using the Uniprot database.

***Human homologs were taken from the ChlamyFP project (**Pazour et al., 2005**).

Segments Q14-R19, K149-K188 and R227-D240 could not be modeled due to poor density in these areas. Similar to what was described above for PACRG, the disconnected fragments of FAP106 were worked out to be part of the same protein based on resolving all the proteins in its environment, the presence of unassigned segments of FAP106 that need to be assigned to density and the density signature agreement with the sequence (*Figure 5—figure supplement 1B*). The model was built and refined as mentioned above.

### FAP126

The density for FAP126, which is mostly disordered, was traced as before to 133 amino acids and ~15 kDa in mass. The density had clear side chains signature, particularly in an area where it appeared to have a Tryptophan residue followed by a Proline, five more amino acids and another Tryptophan residues (*Figure 6—figure supplement 1C*). Doing a blast search against the entire *C. reinhardtii* proteome, in both C- and N-termini directions, using a regular expression matching the pattern above (WPxxxxxW) gives a single hit: FAP126 (Uniprot ID: A8IVJ1). The search was restricted for sequences of lengths between 128 and 147 residues based on the sequence trace. We also applied the same search strategy as FAP106 to identify this protein. The search criteria were: (i) a high stoichiometry number; (ii) a size of ~15 kDa and (iii) no homolog in *Tetrahymena*. In this case, the only protein that satisfied these criteria among high stoichiometry proteins (*Table 2*) was also FAP126. As before, the sequence has matching secondary structure prediction and density signature agreement throughout the entire sequence. The model was modeled and refined as mentioned above.

## Inter-PF angle (lateral curvature) measurement

The inter-PF angle between each PF pair are measured according to *Ichikawa et al. (2017)*.

## Visualization

The maps and models were segmented, coloured and visualized using Chimera (*Pettersen et al., 2004*) and ChimeraX (*Goddard et al., 2018*).

## Mass spectrometry

Sample preparation and mass spectrometry of FAP52 mutant and relative quantification compared to wild type *Chlamydomonas* was done according to *Dai et al. (2019)*. The ratio between the averaged quantitative values from the mass spectrometry (n = 3) and a proteins molecular weight was used to calculate their stoichiometry in the axoneme.

## Transcriptomics analysis

Transcriptomics analysis of PACRG, FAP20, FAP52, FAP126, FAP106, FAP45 and DCX using consensus normalized expression levels for 55 tissue types and seven blood cell types was done according to *Khan et al. (2019)*.

## Acknowledgements

We acknowledge the Facility for Electron Microscopy Research of McGill University where our cryo-EM experiments were conducted, particularly, Drs. Kaustuv Basu and Kelly Sears. We are indebted to Dr. Masahide Kikkawa for sharing the FAP52 knockout *Chlamydomonas* strain. This research was financially supported by Natural Sciences and Engineering Research Council of Canada (RGPIN-2016–04954), Canada Institute of Health Research (CIHR PJT-156354) and the Canada Institute for Advanced Research Arzieli Global Scholars Program to KHB, MI and SK were supported by Japan Society for the Promotion of Science (JSPS) for JSPS Overseas Research Fellowships and JSPS Overseas Challenge Program for Young Researchers, respectively. AAZK is supported by CIHR and the Al Ghurair Foundation for Education. We declare no competing financial interests.

# Additional information

## Funding

| Funder | Grant reference number | Author |
| --- | --- | --- |
| Canadian Institutes of Health Research | PJT-156354 | Khanh Huy Bui |
| Natural Sciences and Engineering Research Council of Canada | RGPIN-2016-04954 | Khanh Huy Bui |
| Natural Sciences and Engineering Research Council of Canada | RGPIN-2018-04813 | Javier Vargas |
| Canadian Institute for Advanced Research | Arzieli Global Scholar Program | Khanh Huy Bui |
| Japan Society for the Promotion of Science | | Muneyoshi Ichikawa Shintaroh Kubo |
| Abdulla Al Ghurair Foundation for Education | Al Ghurair STEM program scholarship | Ahmad Abdelzaher Zaki Khalifa |
| Canada Research Chairs | 950-229792 | Jean-François Trempe |
| Natural Sciences and Engineering Research Council of Canada | RGPIN-2017-04649 | Susanne Bechstedt |

The funders had no role in study design, data collection and interpretation, or the decision to submit the work for publication.

## Author contributions

Ahmad Abdelzaher Zaki Khalifa, Muneyoshi Ichikawa, Data curation, Formal analysis, Investigation; Daniel Dai, Shintaroh Kubo, Corbin Steven Black, Katya Peri, Thomas S McAlear, Formal analysis, Investigation; Simon Veyron, Shun Kai Yang, Formal analysis; Javier Vargas, Software, Formal analysis; Susanne Bechstedt, Formal analysis, Supervision, Funding acquisition; Jean-François Trempe, Formal analysis, Supervision, Funding acquisition, Project administration; Khanh Huy Bui, Conceptualization, Supervision, Investigation, Project administration

## Author ORCIDs

Ahmad Abdelzaher Zaki Khalifa (iD) https://orcid.org/0000-0001-5569-2014
Muneyoshi Ichikawa (iD) https://orcid.org/0000-0002-5921-7699
Daniel Dai (iD) https://orcid.org/0000-0002-9973-0446
Shintaroh Kubo (iD) https://orcid.org/0000-0002-0946-8879
Corbin Steven Black (iD) https://orcid.org/0000-0003-2777-6434
Katya Peri (iD) https://orcid.org/0000-0002-7367-7501
Thomas S McAlear (iD) https://orcid.org/0000-0001-6097-0103
Simon Veyron (iD) https://orcid.org/0000-0002-5533-5268
Shun Kai Yang (iD) https://orcid.org/0000-0002-2363-1441
Javier Vargas (iD) https://orcid.org/0000-0001-7519-6106
Susanne Bechstedt (iD) https://orcid.org/0000-0002-4706-9975
Jean-François Trempe (iD) https://orcid.org/0000-0002-6543-3371
Khanh Huy Bui (iD) https://orcid.org/0000-0003-2814-9889

## Decision letter and Author response

Decision letter https://doi.org/10.7554/eLife.52760.sa1
Author response https://doi.org/10.7554/eLife.52760.sa2

# Additional files

## Supplementary files

• Supplementary file 1. Supplementary Table 1: Significantly reduced or missing proteins in FAP52 compared to WT using relative mass spectrometry quantification.

• Transparent reporting form

## Data availability

Cryo-EM maps have been deposited in EM data bank (EMDB) with accession numbers of EMD-20855 (48-nm averaged Chlamydomonas doublet), EMD-20858 (16-nm averaged Chlamydomonas IJ region) and EMD-20856 (16-nm averaged Tetrahymena IJ region). The model of IJ of Chlamydomonas is available in Protein Data Bank (PDB) with an accession number of PDB: 6VE7. The mass spectrometry is deposited in Dryad (http://doi.org/10.5061/dryad.d51c59zxt).

The following datasets were generated:

| Author(s) | Year | Dataset title | Dataset URL | Database and Identifier |
|---|---|---|---|---|
| Bui KH | 2020 | 48-nm repeat unit of the doublet microtubule from *Chlamydomonas reinhardtii* | http://www.ebi.ac.uk/pdbe/entry/emdb/EMD-20855 | Electron Microscopy Data Bank, EMD-20 855 |
| Bui KH | 2020 | 16-nm averaged Chlamydomonas IJ region | http://www.ebi.ac.uk/pdbe/entry/emdb/EMD-20858 | Electron Microscopy Data Bank, EMD-20 858 |
| Bui KH | 2020 | 16-nm repeat of the doublet microtubule from *Tetrahymena thermophila* | http://www.ebi.ac.uk/pdbe/entry/emdb/EMD-20856 | Electron Microscopy Data Bank, EMD-20 856 |
| Bui KH | 2020 | model of IJ of Chlamydomonas | http://www.rcsb.org/structure/6VE7 | RCSB Protein Data Bank, 6VE7 |
| Khanh Huy Bui | 2020 | MS data for Khalifa et al | http://doi.org/10.5061/dryad.d51c59zxt | Dryad Digital Repository, 10.5061/dryad.d51c59zxt |

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
