## [Decision Letter]

**Acceptance summary:**

This is elegant work where several experimental and computational approaches are combined to identify and locate novel axonemal components. The authors use cryo-electron microscopy to visualize, at high resolution, the structure of proteins that form the inner junction between the A-tubule and the B-tubule of the microtubule doublet in the two species *Chlamydomonas* and *Tetrahymena*. The comparative analysis revealed subtle difference in the inner junction organization between the two species. The authors then use a combination of bioinformatics and mass spectrometry of wild-type and mutant cells to reveal proteins that compose the inner junction and fit them to the cryo-EM structure.

**Decision letter after peer review:**

Thank you for submitting your article "The inner junction complex of the cilia is an interaction hub that involves tubulin post-translational modifications" for consideration by *eLife*. Your article has been reviewed by three peer reviewers, and the evaluation has been overseen by a Reviewing Editor and Vivek Malhotra as the Senior Editor. The reviewers have opted to remain anonymous.

The reviewers have discussed the reviews with one another and the Reviewing Editor has drafted this decision to help you prepare a revised submission.

The reviewers noted that a structure of a *Chlamydomonas* axonemal double was reported by Ma et al. while your work was under consideration. This does not affect the fact that your manuscript is a major contribution to our understanding of the microtubule doublet, which is a fundamental component of every cilium.

Essential revisions:

No experimental revisions are required. The reviewers found parts of the manuscript difficult to follow and suggest work is required to improve the text and figures. The suggestions are listed below.

Reviewer 2:

While much of the structural data in the paper is at high resolution and informative, some of the conclusions drawn from it are not well supported by the data. Words like 'unambiguous' and 'leaves no doubt' are used on several occasions. The fits of the individual proteins into the EM densities are quite convincing, but none of the proteins have been validated structurally using appropriate mutants of tagging of the newly identified IJ proteins.

Another major problem with the manuscript concerns the components that are present in the tomography structures, but absent here. Are the IJ protofilament components not present in *Tetrahymena*, are they missing after the salt wash, or are they at partial occupancy? There are several conflicting sentences throughout the manuscript that make the situation quite confusing:

A) In the third paragraph of subsection “Multiple tether proteins exist at the IJ”, "no IJ PF *Tetrahymena*" Please clarify that this could be due to the biochemical purification used in this study here.

B) Subsection “Multiple tether proteins exist at the IJ” paragraph seven, first sentence: this sentence is contradictory to previous sentences: FAP20/PACRG is present in *Tetrahymena*, but with lower occupancy and also appear to have different occupancy throughout the 96nm repeat.

C) Subsection “PACRG, FAP20, FAP52 and FAP276 form an IJ complex” third sentence: this sentence appears to repetitive to the sentence in the third paragraph of subsection “Multiple tether proteins exist at the IJ”

D) Discussion paragraph two: "in which the IJ PF was washed away…"

Reviewer 3:

My general comment is that the quality of figures should be improved. From some of figures, I am not really convinced that structural models and protein-protein interactions described in the paper are correct. These concerns should be cleared by adding more details in figures and explanations in the main text. Here are details:

1) FAP276 model and interactions. "… the side chain agrees unambiguously with the density signature in this region". Which sequence and density did you use to assign FAP276 to the Y-shaped density? It is unclear from Figure 2—figure supplement 1G and H. Also, the authors described "FAP276 itself forms numerous interact with tubulin". How do they interact? I would like to see their interaction details in a figure.

2) PACRG and FAP20 interactions. Figure 2E and H, do the authors want to show details of protein-protein interactions? Fitness of side chains? It is not clear from these figures. The authors should make magnified views of side-chain / side chain interactions and describe how they exactly interact (e.g. by lines). For instance, in Figure 2H, it is unclear how side chains shown in the figure contribute to the protein-protein interactions between β-tubulin and PACRG and FAP20. Also, where are E432 and F437? These labelled in Figure 2H do not clearly point where they are. Also, the authors described "The PACRG and FAP20 binding interface appears to involve multiple hydrogen bonds…". However, they did not show details of these hydrogen bonds in Figure 3A, B and C. It is particularly important to show how the FAP20-binding loop of PACRG interacts with FAP20, and residues of FAP20 that interact with the loop are conserved (related to Figure 3—figure supplement 1).

3) PACRG model. Figure 2I, there is a missing region in the red model. How did the authors conclude these two red segments are from the same protein?

4) FAP52 interactions. The authors described "FAP52 segment G142-P143 appears to interact with T41 of a-tubulin from PF B9". If residues of these regions have been associated, these interactions should be shown in a figure. Also in Figure 4F, the positions of residues are unclear, nitrogen atoms in R225 are not blue so it was hard to identify the residue. From this figure, I'm not convinced that R225 makes salt bridge with D39.

5) FAP106 model. First paragraph subsection “FAP106 is the Tether loop, consisting of Tether density 1 and 2”, how did the authors conclude that these two separated densities are from the same protein instead of two? Without seeing the connectivity of the densities and knowing the identity of the proteins (upto this point), it is not convincing their conclusion. The loop in *Tetrahymena* looks connected. Did they use this as a support? If so, please explain clearly. Also, subsection “FAP106 is the Tether loop, consisting of Tether density 1 and 2” paragraph four, the authors did not show "the density signature", which must be shown in a figure with the FAP106 model with side chains. Is a "[FHY]xWxxKxx[FHY]" (subsection “FAP106”) motif the density signature? If so, please show the density of the motif and its model in a figure panel.

6) FAP126 model. Subsection “FAP126, a FLTOP homolog, interacts with the tether loop, FAP106” paragraph two, the authors mentioned "WPxxxxxW". Please show a magnified view of this density with a model in a figure panel.

7) FAP106 and FAP126 interactions. Figure 5, the authors should show in a figure panel how side chains of FAP106 and FAP126 interact in details.

---

## [Author Response]

Essential revisions:No experimental revisions are required. The reviewers found parts of the manuscript difficult to follow and suggest work is required to improve the text and figures. The suggestions are listed below.Reviewer 2:While much of the structural data in the paper is at high resolution and informative, some of the conclusions drawn from it are not well supported by the data. Words like 'unambiguous' and 'leaves no doubt' are used on several occasions. The fits of the individual proteins into the EM densities are quite convincing, but none of the proteins have been validated structurally using appropriate mutants of tagging of the newly identified IJ proteins.

We appreciate the comments and we fixed all the texts to make sure it is not over-claimed.

Another major problem with the manuscript concerns the components that are present in the tomography structures, but absent here. Are the IJ protofilament components not present in Tetrahymena, are they missing after the salt wash, or are they at partial occupancy?

Our salt wash definitely affects the stability of specific MIPs. The stability of specific MIPs are not the same in different species. For example, there are many MIPs in the A-tubule of *Chlamydomonas* is not present. For PACRG and FAP20, almost all PACRG and FAP20 are not present in *Tetrahymena*. This should be due to salt wash. Only one pair of PACRG and FAP20 is still there as we reported. This is likely due to a specific interaction within the 96-nm repeat of this pair. We do see some density around this specific pair in *Tetrahymena*, however, since our map is only 48-nm repeating unit, this density is very weak to show.

There are several conflicting sentences throughout the manuscript that make the situation quite confusing:A) In the third paragraph of subsection “Multiple tether proteins exist at the IJ”, "no IJ PF Tetrahymena" Please clarify that this could be due to the biochemical purification used in this study here.B) Subsection “Multiple tether proteins exist at the IJ” paragraph seven, first sentence: this sentence is contradictory to previous sentences: FAP20/PACRG is present in Tetrahymena, but with lower occupancy and also appear to have different occupancy throughout the 96nm repeat.C) Subsection “PACRG, FAP20, FAP52 and FAP276 form an IJ complex” third sentence: this sentence appears to repetitive to the sentence in the third paragraph of subsection “Multiple tether proteins exist at the IJ”D) Discussion paragraph two: "in which the IJ PF was washed away…"

We rewrote the text to clarify this point.

“On the other hand, the IJ region bridging PF B10 and A1 of *Chlamydomonas* doublet remained intact (Figure 1A-D). In the corresponding salt treated *Tetrahymena* doublet map, most of the IJ region bridging PF B10 and A1 is missing (7, 8) (more details later).[…] The presence of the complete IJ PF stabilizes the B-tubule of the *Chlamydomonas* doublet relative to *Tetrahymena*, as evidenced by local resolution measurements (Figure 1—figure supplement 1D).”

Results section:

“The entire IJ filament of PACRG and FAP20 is previously reported missing in the *Tetrahymena* structure, probably due to salt wash and also dialysis (8). However, upon adjusting the threshold value of the surface rendering, we observed one pair of PACRG and FAP20 remaining in the structure (Figure 1—figure supplement 1E) (7). This can be a result of a specific region in the 96-nm repeat of the *Tetrahymena* doublet that has extra interactions to prevent their detachment during sample preparation.”

“Since the B-tubule is flexible in *Tetrahymena*, the resolution in the IJ area was significantly lower than that of *Chlamydomonas* (Figure 1—figure supplement 1D).”

Discussion

“In our *Tetrahymena* doublet, in which most of the IJ PF was washed away, even with the presence of FAP52 and FAP106, the doublet is still flexible which can be seen by the lower resolution of the B-tubule compared to the A-tubule (Figure 1—figure supplement 1D).”

Reviewer 3:My general comment is that the quality of figures should be improved. From some of figures, I am not really convinced that structural models and protein-protein interactions described in the paper are correct. These concerns should be cleared by adding more details in figures and explanations in the main text. Here are details:1) FAP276 model and interactions. "… the side chain agrees unambiguously with the density signature in this region". Which sequence and density did you use to assign FAP276 to the Y-shaped density? It is unclear from Figure 2—figure supplement 1G and H. Also, the authors described "FAP276 itself forms numerous interact with tubulin". How do they interact? I would like to see their interaction details in a figure.

We removed the word “unambiguously” to make sure we are not overclaiming. We added details about sequence for all proteins now in Materials and methods now.

Also, to illustrate the interaction from FAP276 with tubulins, we replaced Figure 2C with a new one, showing how FAP276 insert into the tubulin lattice. In order for not overinterpreting the interaction, we wrote now “FAP276 itself forms different contacts with tubulin with both of its N- and C-termini, thus it provides strong anchorage for FAP52 to the tubulin lattice (Figure 2C).”

2) PACRG and FAP20 interactions. Figure 2E and H, do the authors want to show details of protein-protein interactions? Fitness of side chains? It is not clear from these figures. The authors should make magnified views of side-chain / side chain interactions and describe how they exactly interact (e.g. by lines).

We improved Figure 2E, F and H for this purpose. The new Figure 2E is swapped with Figure 2G. Also, with the purpose to not overinterpret the map at this resolution, we highlighted the H136 as important residues for interaction with B10’s tubulin since H136 is conserved from different species (Khan et al., 2019) while other residues in this loop are not conserved. We also reflected this in the text.

For instance, in Figure 2H, it is unclear how side chains shown in the figure contribute to the protein-protein interactions between β-tubulin and PACRG and FAP20. Also, where are E432 and F437?

We labelled the tubulin now more clearly. Also, to not overinterpret, indicate the potential residues in the interaction in the figures and said it in the text.

These labelled in Figure 2H do not clearly point where they are. Also, the authors described "The PACRG and FAP20 binding interface appears to involve multiple hydrogen bonds…". However, they did not show details of these hydrogen bonds in Figure 3A, B and C. It is particularly important to show how the FAP20-binding loop of PACRG interacts with FAP20, and residues of FAP20 that interact with the loop are conserved (related to Figure 3—figure supplement 1).

To illustrate the interaction of FAP20 and PACRG, we now include a new Figure 3B showing the detail interaction region. We also highlight the region of FAP20 interaction with PACRG in its sequence alignment in Figure 3—figure supplement 1. Here is what we write now about this interaction in the text:

“The PACRG and FAP20 binding interface involves complementary surface charges, suggesting a specific and strong interaction (Figure 3B-D). The loop N225-I260 of PACRG forms a β sheet stacking like interactions with the strand H33-R36 of FAP20 (Figure 3B). In addition, residue Q264 of PACRG forms a hydrogen bond with residue T38 of FAP20.”

3) PACRG model. Figure 2I, there is a missing region in the red model. How did the authors conclude these two red segments are from the same protein?

They are part of the same protein because 1) all the surrounding densities have been assigned to the α-β-tubulin dimers, 2) There is a 101 long segment of the PACRG N-terminus that is still unassigned to density and 3) the density signature downstream and upstream of the gap matches the sequence identity of the PACRG N-terminus as shown in Figure 2—figure supplement 1B.

We included this in the Materials and methods.

4) FAP52 interactions. The authors described "FAP52 segment G142-P143 appears to interact with T41 of a-tubulin from PF B9". If residues of these regions have been associated, these interactions should be shown in a figure. Also in Figure 4F, the positions of residues are unclear, nitrogen atoms in R225 are not blue so it was hard to identify the residue. From this figure, I'm not convinced that R225 makes salt bridge with D39.

We improved Figure 4F and G to reflect the suggestions. We removed the text above and wrote the new text as follows. We did not observe the side chain for D39 or T41 due to low resolution in this region. Therefore, we can only speculate about the interaction based on proximity.

We rewrote as follow:

“The density of the α-K40 loop of PF B10 is more complete, allows us to hypothesize that residue K229 of FAP52 interacts with the backbone of D39 of α-tubulin from PF B10. It is also possible that there is a hydrophobic interaction between L226 and I42 (Figure 4E). Residues R225 of FAP52 and D39 of α-tubulin are in an environment where they could form an interaction. However, the side chain densities for both residues are not well-resolved. At the second tubulin contact point, FAP52 segment G142-P143 is in the close proximity of the α-K40 loop from PF B9 (Figure 4B). In the lower region of FAP52, the loop V268-L279 is in close proximity with the N-terminus of PACRG (Figure 4B).”

5) FAP106 model. First paragraph subsection “FAP106 is the Tether loop, consisting of Tether density 1 and 2”, how did the authors conclude that these two separated densities are from the same protein instead of two? Without seeing the connectivity of the densities and knowing the identity of the proteins (upto this point), it is not convincing their conclusion. The loop in Tetrahymena looks connected. Did they use this as a support? If so, please explain clearly. Also, subsection “FAP106 is the Tether loop, consisting of Tether density 1 and 2” paragraph four, the authors did not show "the density signature", which must be shown in a figure with the FAP106 model with side chains. Is a "[FHY]xWxxKxx[FHY]" (subsection “FAP106”) motif the density signature? If so, please show the density of the motif and its model in a figure panel.

We used the Tetra as the support for the connectivity of this region. We reflected this now in the Materials and methods.

We added the motif "[FHY]xWxxKxx[FHY]" of the density signature as Figure 5—figure supplement 1A now.

6) FAP126 model. Subsection “FAP126, a FLTOP homolog, interacts with the tether loop, FAP106” paragraph two, the authors mentioned "WPxxxxxW". Please show a magnified view of this density with a model in a figure panel.

We added Figure 6—figure supplement 1C to show the density of the motif to search for FAP126 identity. Also label the corresponding sequence in Figure 6—figure supplement 1A.

7) FAP106 and FAP126 interactions. Figure 5, the authors should show in a figure panel how side chains of FAP106 and FAP126 interact in details.

We improve Figure 6B to highlight the interaction between FAP106 and FAP126.